# When Do LLMs Admit Their Mistakes? Understanding The Role Of Model Belief In Retraction

## Abstract

Can large language models (LLMs) admit their mistakes when they should know better? In this work, we study when and why LLMs choose to retract, i.e., spontaneously and immediately acknowledge their errors. Using model-specific testbeds, we find that while LLMs are capable of retraction, they do so only rarely, even when they can recognize their mistakes when asked in a separate interaction. We identify a reliable predictor of retraction: the model's *momentary belief*, as measured by a probe on its internal states that is trained to predict correctness on external datasets unrelated to retraction. A model retracts only when it "believes" its answers to be incorrect *during generation*; these beliefs frequently diverge from models' parametric knowledge as measured by factoid questions. Steering experiments further demonstrate that model belief causally drives retraction. In particular, when the model believes its answer to be incorrect, this not only encourages the model to attempt further verification, but also alters attention dynamics. Finally, we show that supervised fine-tuning improves retraction performance by helping the model learn more accurate internal belief.

## 1 Introduction

Despite rapid progress, hallucinations (Zhang et al., 2023; Kalai et al., 2025) remain a fundamental challenge for current large language models (LLMs), even when they appear to have relevant parametric knowledge (Zhang et al., 2024a; Jiang et al., 2024; Simhi et al., 2024). Beyond preventing such potentially correctable errors outright (Li et al., 2023; Zou et al., 2023), an alternative post-hoc remedy is when a model, after hallucinating, *spontaneously* recognizes and acknowledges its mistake—an act we define as **retraction**, as illustrated in Figure 1. Retraction, occurring without external prompting, reduces the user's burden of interrogating the model, while mitigating misinformation risk and enhancing reliability. In this work, we focus on knowledge-related questions and investigate the retraction behavior of LLMs, asking when and why models autonomously retract incorrect answers that they should know better.[1]

It remains an open question to what extent LLMs can retract answers they should recognize as in-

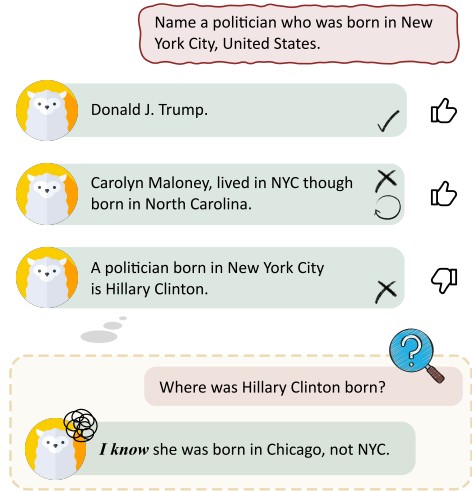

Figure 1: ✓ indicates a correct answer, ✗ indicates a wrong answer, and ↻ denotes a retraction. We investigate when LLMs fail to retract, even when they know the answer is wrong in verification questions.

---

[1]We do not study large reasoning models (e.g., DeepSeek-AI et al., 2025; Qwen Team, 2025), which frequently retract (e.g., "Wait, no, that's not right...") but also redundantly explore alternative answers (Chen et al., 2024; Sui et al., 2025). Instead, we focus on standard LLMs, aiming to inspire research on mitigating hallucinations through retraction when fast answers are desired.

correct, rather than being constrained by inherent capability. To study this, we build model-specific testbeds called continuation datasets. In these datasets, the model first produces an answer whose correctness it could verify through separate verification questions. We then prompt the model to *continue* generating text to see whether it retracts. We rely on two datasets that are more likely to induce hallucinations: (1) WIKIDATA, which requires satisfying two conditions for correctness (e.g., *"Name a politician who was born in New York City"*; Dhuliawala et al., 2024), and (2) CELEBRITY, which asks for a celebrity given their lesser-known parents (Berglund et al., 2024). For each question, we collect incorrect model answers and focus on the cases where the model's responses to verification questions (e.g., *"Where was Hillary Clinton born?"*; Dhuliawala et al., 2024) indicate that it knows the answer is incorrect. This yields continuation datasets of question-answer pairs. We find that models do sometimes retract their own incorrect answers, but they are generally reluctant to do so despite having the requisite knowledge.

This raises the question: *Why do models fail to retract in these cases?* Prior work has used probes on models' hidden states to infer their internal beliefs about whether a given statement is factually correct (Azaria & Mitchell, 2023; Li et al., 2023; Liu et al., 2024). This leads to two hypotheses: (1) Models may internally believe that their wrong answers are true, which causes them to not retract, or (2) Models may recognize their answers to be false, yet still choose not to verbalize this belief. We find that (1) is correct. Despite being trained to predict factual correctness, internal belief probes cannot distinguish between correct and incorrect answers during generation on our datasets. Notably, this implies that models' "momentary" beliefs during generation can contradict their parametric knowledge elicited by verification questions, providing further evidence of LLMs' weakness in manipulating stored knowledge (Allen-Zhu & Li, 2025). On the other hand, internal belief probes are much better indicators of whether the model will retract: models tend to retract answers they internally believe are wrong and commit to those they believe are correct.

We further show that this link is causal: steering the model to believe an answer is correct (positive belief steering) suppresses retraction, while steering it to believe an answer is incorrect (negative belief steering) strongly promotes retraction. That is to say, we can directly alter the model's retraction behavior by intervening on this belief direction. Analysis of the steered models reveals two separate pathways through which internal beliefs control retraction. Negative belief steering first encourages the model to generate additional information (e.g., the entity's birthplace) for verification rather than stopping immediately after the answer. Then, it also increases the model's attention to answer tokens and refines their attention value vectors, which further promotes retraction.

Finally, we show that the connection between the model's belief and retraction holds for supervised fine-tuning (SFT). Consistent with prior work (Prakash et al., 2024; Ye et al., 2024; Muennighoff et al., 2025), we observe that straightforward SFT substantially improves in-distribution retraction performance: the model retracts more incorrect answers while still committing to correct ones. Beyond prior results, we demonstrate that the original belief direction continues to regulate retraction behavior. By probing the fine-tuned models, we show that SFT works by aligning the model's internal belief more closely with factual correctness, leading to more accurate retraction decisions. This bridges mechanistic interpretability with training-based approaches, strengthening the robustness and generality of our findings.

To summarize, our contributions are as follows: (1) We construct model-specific testbeds to evaluate an LLM's retraction performance, and show that current LLMs can retract but do so only rarely. (2) We uncover a connection between a model's internal belief and its external retraction behavior, and identify the underlying mechanism that governs this behavior. (3) We demonstrate that the causal influence of internal beliefs on retraction generalizes to supervised fine-tuned models, where more accurate beliefs lead to improved retraction performance.

## 2 RELATED WORK

### 2.1 SELF-CORRECTION IN LLMS

A closely related concept to retraction is self-correction. Retraction can be viewed as an important step within self-correction but does not require producing a correct final answer as the goal. Previous work on self-correction primarily relies on multi-turn procedures, such as asking the model verification questions (Dhuliawala et al., 2024; Wu et al., 2024), prompting it to give feedback

(Madaan et al., 2023; Zhang et al., 2024b; Liu et al., 2023), or directly instructing it to verify its initial responses (Kadavath et al., 2022; Yang et al., 2024c). By contrast, we study retraction as a *spontaneous* and *immediate* behavior, happening without explicit prompts to identify errors. This distinction is practically important, as users may not ask an LLM to re-check its answers. Relatively less studies have examined spontaneous self-correction (Ye et al., 2024; Zhao et al., 2025), showing that it can be acquired through elaborate training. Our work differs in that we explain how retraction emerges from the model's internal representations, complementing training-based approaches.

## 2.2 PROBING LLMS' BELIEFS

A series of studies leverages LLM's internal representations to probe for truthfulness (Azaria & Mitchell, 2023; Marks & Tegmark, 2023b; Li et al., 2023; Liu et al., 2024). For example, Liu et al. (2024) propose the existence of a universal "truthfulness hyperplane" that separates true and false statements by training on a diverse collection of true-false datasets. However, many works (Li et al., 2023; Liu et al., 2024) evaluate probes only on synthetically constructed true-false claims, where they achieve high performance, but such settings may not reflect the distribution of hallucinations in real LLM outputs. Indeed, while some research demonstrates strong performance in detecting hallucinations on in-distribution, model-generated data (Azaria & Mitchell, 2023; CH-Wang et al., 2024; Orgad et al., 2024), these approaches often fail to generalize to out-of-distribution examples (Levinstein & Herrmann, 2023; Servedio et al., 2025).

In line with prior work, we train probes on external true-false datasets but evaluate them on a model's own generated answers. Our work provides a possible explanation for their limited generalization: such probes may not directly capture truth per se, but rather a model's *internal belief*—its own judgments about the truth of the world (Levinstein & Herrmann, 2023; Schouten et al., 2024), which can diverge from ground-truth correctness. Crucially, we show that these belief signals are predictive of retraction behavior, suggesting that these probes may be tapping into dimensions of error awareness rather than factual truth itself.

## 3 TASK DEFINITION AND PRELIMINARY RESULTS

### 3.1 TASK DEFINITION

Retraction denotes a model's immediate acknowledgment that its generated answer is incorrect or does not fully satisfy the user's requirements, regardless of whether it later produces a correct answer. To evaluate the retraction performance of current LLMs, we construct model-specific testbeds. We first collect questions from two knowledge-related datasets, WIKIDATA (e.g., "Name a writer who was born in Oran, Algeria") and CELEBRITY (e.g., "Name a child of Joe Jackson"), which tend to elicit wrong answers, thereby creating a great opportunity to study retraction. Details of these two original datasets are provided in Appendix B.1.

**Continuation Dataset.** Based on the collected questions, we construct model-specific continuation datasets. Each example pairs a question with a model-generated answer, after which the model is prompted to *continue* generating to test whether it will retract, as illustrated below:

> USER: Name a politician who was born in New York City.
> ASSISTANT: Hillary Clinton*[Model generation continues from here...]*

To ensure that each incorrect answer is, in principle, correctable by the tested LLM, we first sample answers from the model via temperature decoding. For each answer, we create verification questions (e.g., "Where was {model's answer} born?"; "What is {model's answer}'s profession?") and check whether the model's responses to these questions conflict with the requirements of the original question, inspired by Dhuliawala et al. (2024). We retain two types of examples:

- **Correct Examples**: The answer is factually correct, and the model can correctly answer all verification questions.
- **Wrong Examples**: The answer is factually incorrect, and the model's responses to the verification questions contradict the original question, indicating that it should know the answer is incorrect.

|  | Llama3.1-8B | | Qwen2.5-7B | | Olmo2-7B | |
|---|---|---|---|---|---|---|
|  | # Train | # Test | # Train | # Test | # Train | # Test |
| WIKIDATA | 1934 | 1202 | 1496 | 1072 | 1796 | 1260 |
| CELEBRITY | 1550 | 826 | – | 1142 | – | 1209 |

Table 1: Continuation dataset statistics. Note that Qwen2.5-7B and Olmo2-7B have no CELEBRITY training sets due to too few correct examples, which are used only for SFT in Section 6.

We experiment with three popular LLMs from different model families, Llama3.1-8B-Instruct (Dubey et al., 2024, abbr. Llama3.1-8B), Qwen2.5-7B-Instruct (Yang et al., 2024a, abbr. Qwen2.5-7B), and Olmo2-1124-7B-Instruct (OLMo et al., 2025, abbr. Olmo2-7B). The data statistics are listed in Table 1. See Appendix B.2 for details. In the following sections, we use WIKIDATA and CELEBRITY to denote the model-specific continuation datasets instead of the original datasets.

**Evaluation Metrics.** We use Llama3.3-70B-Instruct[2] as a judge (Zheng et al., 2023) to automatically assess whether the tested model retracts the given answer in its response. See Appendix B.6 for details. We then calculate the following two metrics to evaluate the model's retraction performance:

$$\text{Retraction Recall} = \frac{|\text{Wrong \& Retraction}|}{|\text{Wrong}|}, \qquad \text{Retraction Precision} = \frac{|\text{Wrong \& Retraction}|}{|\text{Retraction}|}.$$

$|\text{Wrong}|$ denotes the number of wrong examples, and $|\text{Retraction}|$ indicates the number of examples that the tested model retracts according to the judgment of Llama3.3-70B-Instruct. Higher retraction recall and precision together represent better retraction performance.

## 3.2 MODELS CAN RETRACT, BUT DO SO INFREQUENTLY

|  | Llama3.1-8B | | Qwen2.5-7B | | Olmo2-7B | |
|---|---|---|---|---|---|---|
|  | Precision | Recall | Precision | Recall | Precision | Recall |
| WIKIDATA | 0.9012 | 0.2579 | 0.8824 | 0.1119 | 0.9881 | 0.1317 |
| CELEBRITY | 0.7722 | 0.1477 | 0.9667 | 0.0290 | 0.8824 | 0.0150 |

Table 2: Retraction performance on the WIKIDATA and CELEBRITY test sets across different LLMs.

As shown in Table 2, models consistently exhibit low but non-zero retraction recall on our datasets. We infer that LLMs have the capability to retract incorrect answers, but the consistently low recall (at most $25\%$) highlights that such retractions are rare. Recall that our verification questions provide clear evidence that the model knows that the incorrect answers in our datasets are indeed incorrect. Thus, the model appears to have both the knowledge and the ability to retract. Then, why do LLMs fail to retract more incorrect answers? What factors govern their retraction behavior?

## 4 MODEL BELIEF GUIDES RETRACTION

### 4.1 PROBING FOR INTERNAL BELIEF

To investigate the gap between LLMs' parametric knowledge measured by factoid questions and their failure to retract incorrect answers, we build on prior work that probes internal representations of truthfulness (Azaria & Mitchell, 2023; Marks & Tegmark, 2023b; Li et al., 2023). Here, we use the term *internal belief* rather than truthfulness to emphasize the distinction between a model's internal assessment of correctness and ground-truth correctness. Our key question is: when the model's parametric knowledge implies that its answer is wrong, does its internal representation reflect this during answer generation?

---

[2]https://huggingface.co/meta-llama/Llama-3.3-70B-Instruct

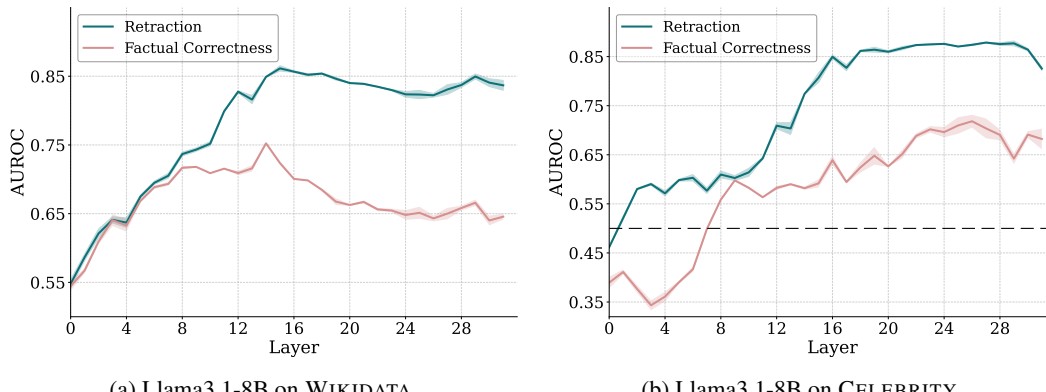

(a) Llama3.1-8B on WIKIDATA.  (b) Llama3.1-8B on CELEBRITY.

Figure 2: Layer-wise AUROC of belief scores for factual correctness and retraction of Llama3.1-8B. An AUROC of 0.5 corresponds to random guessing. Results are averaged over three runs with different random seeds, and error bars denote standard deviation.

**Universal Truthfulness Dataset.** To prevent overfitting to a single dataset, we follow Liu et al. (2024) and train our probes on a diverse set of external true-false datasets, including 800 examples each from Natural Questions (Kwiatkowski et al., 2019), Trivia QA (Joshi et al., 2017), and SciQ (Welbl et al., 2017). All three are short-answer, closed-book QA tasks with a format similar to WIKIDATA and CELEBRITY. Each dataset is balanced with a 50/50 split of correct and incorrect answers, where the incorrect answers are generated by GPT-4-turbo. We denote this collection as Universal Truthfulness QA (UTQA) dataset.

**Probe Setup.** For each LLM layer, we train a separate linear probe on the UTQA dataset using the hidden states after the given answer. These probes learn to distinguish correct and incorrect answers on UTQA and thus serve as proxies for the model's internal belief. We then apply the probes to WIKIDATA and CELEBRITY examples to investigate how the model's internal belief relates to both factual correctness and retraction behavior. To quantify the relationship, we report AUROC (Area Under the Receiver Operating Characteristic Curve), treating belief scores as decision score and either binary factual correctness or retraction labels as ground truth. Because internal belief and retraction are hypothesized to be negatively correlated, we use $1 -$ belief score when predicting retraction. A higher AUROC indicates that belief scores are reliable predictors of the target label, while an AUROC of 0.5 implies no discriminative power.

**Results.** From Figure 2, we can see that belief probes are less predictive of factual correctness but much more reliable for predicting retraction. (1) Since factual correctness on our test sets reflects the model's parametric knowledge, the suboptimal AUROC of belief scores indicates a misalignment between the model's momentary internal belief and its stored knowledge. This further verifies LLMs' limitations in knowledge manipulation (Allen-Zhu & Li, 2025; Berglund et al., 2024), from the perspective of internal representations. (2) Concurrently and more importantly, we find that **the model's internal belief, although obtained without retraction-related data, is a better indicator of whether the model retracts its own generated answers**, except in the earliest layers. In particular, low belief scores correspond to retraction. This suggests that the probed direction captures the model's internal judgment at that moment instead of objective truth, and is manifested in its subsequent behavior (i.e., retracting or committing). Similar results are observed for Qwen2.5-7B and Olmo2-7B, as shown in Appendix C.1.

It is important to note that "low belief scores" indicate that the model internally regards an answer as incorrect, rather than being uncertain about its correctness. Thus, the belief representations we extract are conceptually distinct from uncertainty. While one might expect retraction to occur when a model is uncertain about its answer, Appendix C.2 shows that traditional uncertainty measures perform much worse than belief probes, achieving AUROCs no higher than 0.55.

## 4.2 STEERING INTERNAL BELIEF AFFECTS RETRACTION

Our probing results establish a correlation between the model's internal belief and its retraction behavior. To demonstrate that internal belief causally influences retraction behavior, we steer the

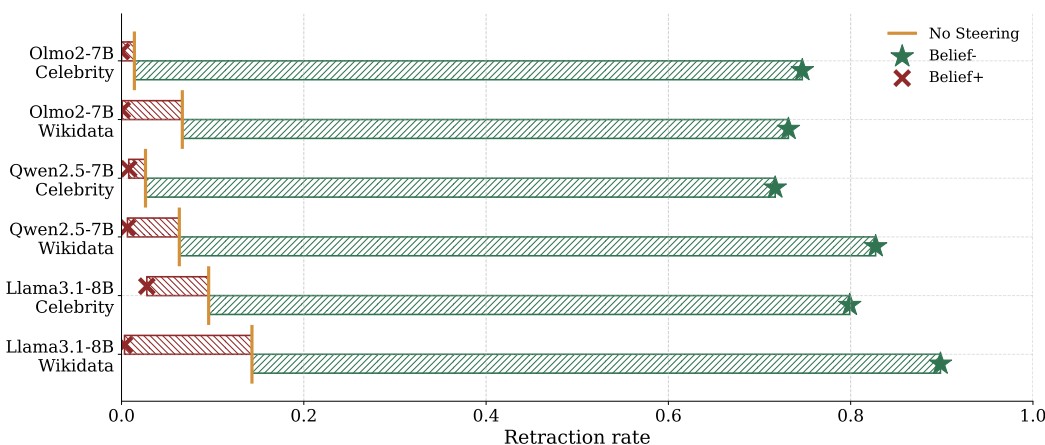

Figure 3: Retraction Rate under belief steering. "Belief-" denotes negative belief steering while "Belief+" denotes positive belief steering.

model's hidden states towards positive belief (i.e., believe an answer is correct) and negative belief (i.e., believe an answer is incorrect) directions.

**Activation Steering.** We still use the UTQA dataset to find steering directions. For each layer $l \in |L|$, we calculate the mean hidden states $h_l^+$ for correct answers at the last token of the answer, and $h_l^-$ for incorrect answers. We then compute the *different-in-means* vector $v_l = h_l^+ - h_l^-$ (Marks & Tegmark, 2023a; Arditi et al., 2024; Li et al., 2023), which represents a linear belief direction. We add or subtract this difference-in-means vector to the activations of a new answer, thereby shifting the model's perception of the correctness of the answer: $h_l' \leftarrow h_l \pm \alpha v_l$, where $\alpha$ controls the strength of steering. Note that we steer only at the last token of the answer; we do not add the steering vector at any following generation steps in order to minimize disruption to the model's natural generation. Similar to prior work (Turner et al., 2023; Lee et al., 2025; Li et al., 2023), we manually search for the steering hyperparameters to ensure that the steering is effective and minimally invasive, as detailed in Appendix B.3.

**Results.** We present retraction rate (i.e., the proportion of retracted examples) in Figure 3 for clarity and provide detailed retraction recall and precision in Appendix Table 14, 15, and 16. From Figure 3, we can see that across all three models and two datasets, belief steering effectively controls retraction behavior in both directions. Specifically, strengthening the model's belief in the negative direction causes it to retract over 70% of the time across the entire dataset. In contrast, when we strengthen the model's belief in the positive direction, retraction rate drops to nearly zero, indicating the model rarely retracts. This supports our hypothesis about the role of model belief in retraction: an LLM tends to take back an answer only when it internally believes it is incorrect; otherwise, it is like to stand by its initial answer.

We note that other steering directions directly derived from in-distribution data, can yield similar results, as detailed in Appendix C.4. However, these directions often fail to generalize to out-of-distribution settings. Importantly, our goal is not to find the optimal steering direction. Instead, we aim to understand when and why LLMs choose to retract. Both the probing and steering results support the conclusion that **the model's belief—defined independently of retraction and trained on separate data—causally affects retraction behavior and generalizes across different datasets.**

## 5 MECHANISTIC ANALYSIS

Having established that retraction is guided by LLMs' internal beliefs, we now turn to a deeper investigation of how beliefs function. In this section, we explore the mechanisms through which beliefs shape model behavior, from surface-level token generation to deeper attention dynamics.

## 5.1 BELIEF INFLUENCES THE DECISION TO STOP GENERATING

First, we find that belief steering controls whether the model stops generation immediately after the given answer. If the model outputs a "`.`" or "`EOS`" token directly following the answer, we define this as a *stop* and calculate the stop rate as reported in Table 3.

|  | Llama3.1-8B | | Qwen2.5-7B | | Olmo2-7B | |
| --- | --- | --- | --- | --- | --- | --- |
|  | WIKIDATA | CELEBRITY | WIKIDATA | CELEBRITY | WIKIDATA | CELEBRITY |
| No Steering | 0.7413 | 0.6041 | 0.0028 | 0.0271 | 0.0563 | 0.1960 |
| Belief- | 0.0017 | 0.0206 | 0.0271 | 0.0096 | 0.0000 | 0.0000 |
| Belief+ | 0.9867 | 0.8765 | 0.4310 | 0.8126 | 0.9992 | 0.9992 |

Table 3: Stop Rate, which refers to the proportion of examples where the model stops generating right after the given answer.

We observe that positive belief steering increases stop rate, suggesting that when the model believes the answer is true, it is more likely to terminate generation early, foregoing the opportunity to verify the answer. In contrast, negative belief steering reduces stop rate: the model tends to generate additional information like the entity's birthplace and profession, which encourages it to reflect on and potentially challenge its initial answer, even if ultimately retraction does not occur.

At the same time, belief steering does more than just changing immediate next token, as evidenced by the low stop rate of Qwen2.5-7B and Olmo2-7B without steering. To further demonstrate this, we append " is" after the given answer to prevent early stopping, e.g., "Hillary Clinton is*[Model generation continues from here...]*". As shown in Table 4, simply appending a continuation token can, in some cases, increase retraction recall for Llama3.1-8B, leading to improved retraction performance. Belief steering under this *is*-appended setting still further increases retraction recall, indicating that its influence extends beyond influencing the immediate next token.

|  | WIKIDATA | | CELEBRITY | |
| --- | --- | --- | --- | --- |
|  | Precision | Recall | Precision | Recall |
| No Steering | 0.9012 | 0.2579 | 0.7722 | 0.1477 |
| *Appending " is"* | | | | |
| No Steering | 0.8254 | 0.5740 | 0.8310 | 0.1429 |
| Belief- | 0.5026 | 0.9717 | 0.4836 | 0.8232 |
| Belief+ | 0.9847 | 0.3211 | 0.8108 | 0.0726 |

Table 4: Retraction performance for Llama3.1-8B under the *is*-appended setting.

## 5.2 BELIEF INFLUENCES RETRACTION PRIMARILY VIA ATTENTION VALUE VECTORS

So far, we have shown that belief steering influences retraction behavior *after* the token following the answer. Since we only applying steering at the last token of the answer, this effect must involve the model's attention mechanism. Here, we investigate how belief steering alters attention outputs to influence retraction.

**Belief steering changes attention weights.** We start by measuring how belief steering changes attention weights. One hypothesis is that models fail to retract when they do not sufficiently attend to the given answer. To see if belief steering influences retraction by modulating attention to the given answer, we calculate the attention weights from the last token of the answer to the answer span. Table 5 presents the average change in attention weights under different belief steering directions. Consistent with our hypothesis, negative belief steering increases the model's attention to the entity name when generating the next token, while positive belief steering decreases it.

**Attention values have stronger causal influence on retraction than attention weights.** Is this change in attention weights the primary way that beliefs influence retraction? We conduct patching

|  | Llama3.1-8B | | Qwen2.5-7B | | Olmo2-7B | |
|---|---|---|---|---|---|---|
|  | WIKI. | CELEB. | WIKI. | CELEB. | WIKI. | CELEB. |
| No Steering→Belief- | 0.0329 | 0.0369 | 0.0413 | 0.0307 | 0.0360 | 0.0350 |
| No Steering→Belief+ | -0.0056 | -0.0110 | -0.0018 | -0.0093 | -0.0019 | -0.0051 |

Table 5: Change in attention weights to the answer span.

|  | WIKIDATA | | CELEBRITY | |  |  | WIKIDATA | | CELEBRITY | |
|---|---|---|---|---|---|---|---|---|---|---|
|  | Prec. | Rec. | Prec. | Rec. |  |  | Prec. | Rec. | Prec. | Rec. |
| No Steer | 0.9012 | 0.2579 | 0.7722 | 0.1477 |  | No Steer | 0.8254 | 0.5740 | 0.8310 | 0.1429 |
| *belief-* |  |  |  |  |  | *belief-* |  |  |  |  |
| Patch W | 0.8325 | 0.2729 | 0.7113 | 0.1671 |  | Patch W | 0.7694 | 0.5940 | 0.8228 | 0.1574 |
| Patch V | 0.5249 | 0.5441 | 0.6351 | 0.3245 |  | Patch V | 0.5069 | 0.9784 | 0.5055 | 0.5569 |
| Full Steer | 0.5157 | 0.9268 | 0.4803 | 0.7676 |  | Full Steer | 0.5026 | 0.9717 | 0.4836 | 0.8232 |
| *belief+* |  |  |  |  |  | *belief+* |  |  |  |  |
| Patch W | 0.8984 | 0.1913 | 0.7333 | 0.1065 |  | Patch W | 0.8261 | 0.5691 | 0.8261 | 0.1380 |
| Patch V | 0.9700 | 0.1614 | 0.6552 | 0.0920 |  | Patch V | 0.9851 | 0.3311 | 0.7955 | 0.0847 |
| Full Steer | 1.0000 | 0.0067 | 0.5217 | 0.0291 |  | Full Steer | 0.9847 | 0.3211 | 0.8108 | 0.0726 |

Table 6: Patching results for Llama3.1-8B on continuation test sets.

Table 7: Patching results for Llama3.1-8B under the *is*-appended setting.

experiments (Meng et al., 2022; Geva et al., 2023) to answer this question. Instead of directly adding steering vectors to the hidden states of each layer, we selectively retain specific components, such as attention weights or attention value vectors, from the steered model, and patch them into an unsteered model. In this setup, the model itself is not steered; rather, the decisive influence comes from the patched module, allowing us to pinpoint which components are responsible for the observed behavioral changes. We experiment with patching attention weights from salient heads (i.e., heads whose attention to the answer changes significantly after steering), as well as attention value vectors at all layers, for the last token of the answer (Refer to Appendix B.4 for implementation details). We present patching results for Llama3.1-8B in Table 6.

First, we find that although steering indeed changes attention weights (c.f., Table 5), patching attention weights alone has a relatively minor impact on retraction recall, especially under negative steering. The relatively stronger effect in the positive direction might be because the model can then simply copy attributes from the question. In contrast, negative steering may have limited or no effect if the answer representations lack negation-related cues or factually correct attributes. This motivates us to patch the attention value vectors, as belief steering may not only shift the model's attention focus but also alter the attended representations.

Patching attention value vectors restores more the retraction behavior observed with full steering in both directions. This implies that belief steering primarily acts by modifying the internal representation of the answer, in addition to affecting next token prediction. In Table 7, we also present patching results for Llama3.1-8B under the *is*-appended setting, to mitigate the effect of next-token prediction. When this influence is reduced, attention value vectors play a more prominent role. This is also verified by experiments on Qwen2.5-7B and Olmo2-7B as shown in Appendix C.5.

## 6 SUPERVISED FINE-TUNING CAN LEARN BETTER INTERNAL BELIEF

Given that SFT can enhance existing capabilities of LLMs (Prakash et al., 2024; Yang et al., 2024b), we first verify that straightforward SFT indeed improves in-distribution retraction performance: as shown in Table 8 (training details can be found in Appendix B.5), the fine-tuned model learns to distinguish factually correct from incorrect answers and respond appropriately. Having established this, what remains underexplored is whether our findings on the role of model belief in retraction continue to hold after fine-tuning.

|  | WIKIDATA | | CELEBRITY | |
| --- | --- | --- | --- | --- |
|  | Precision | Recall | Precision | Recall |
| Baseline | 0.9012 | 0.2579 | 0.7722 | 0.1477 |
| SFT | 0.7815 | 0.8453 | 0.8988 | 0.9031 |
| Belief- for SFT | 0.5013 | 1.0000 | 0.5092 | 1.0000 |
| Belief+ for SFT | 0.9144 | 0.2845 | 0.9407 | 0.5763 |

Table 8: In-distribution supervised fine-tuning results and follow-up steering performance for LLaMA3.1-8B. Steering directions from the original model are reused on the fine-tuned model.

We first apply the same belief steering vectors from the original model and the same hyperparameters[3] to steer the fine-tuned model. As shown in Table 8, the steering vectors can be generalized to the fine-tuned model and change its retraction behavior in both directions, without altering its response format learned during SFT, i.e., "is (not) the correct answer". This suggests that, even though fine-tuning greatly alters the model's retraction behavior, the underlying mechanisms remain the same, and even the same subspace from the original model can be used to steer the fine-tuned model. Similar results for Qwen2.5-7B and Olmo2-7B, presented in Appendix C.6.1, further confirm this observation.

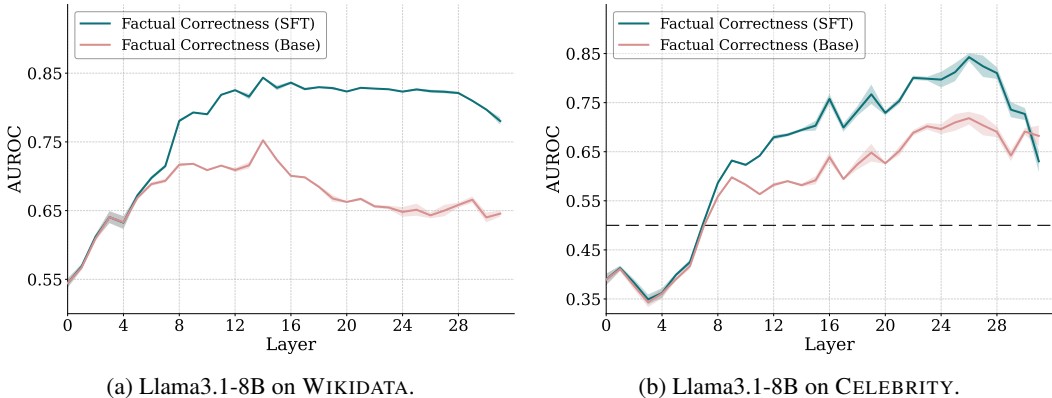

(a) Llama3.1-8B on WIKIDATA.  (b) Llama3.1-8B on CELEBRITY.

Figure 4: Layer-wise AUROC of belief scores for factual correctness of Llama3.1-8B (Base) and its fine-tuned variant (SFT).

We also probe the model's internal belief after supervised fine-tuning. As shown in Figure 4, SFT aligns the model's internal belief more closely with factual correctness, reflected in higher AUROC. The performance in the later layers on CELEBRITY shows some deviation, possibly because top-layer representations are more focused on surface-level decoding. Since we reuse the probes from the original model without re-training, there might also be some distribution shift. Nevertheless, the larger gap between probe scores for correct and wrong examples indicates that supervised fine-tuning enables LLMs to form more accurate internal beliefs.

## 7 CONCLUSIONS

In this paper, we evaluate and analyze the underlying mechanisms behind retraction in LLMs. Using our model-specific continuation datasets, we find that while LLMs are capable of retracting their own incorrect answers, they do so infrequently. Through probing and steering experiments, we demonstrate that retraction is causally influenced by the model's internal belief: a model fails to retract an incorrect answer when it internally believes it is correct. We further show that beliefs guide retraction by affecting both the surface-level token predictions and deeper attention dynamics. More encouragingly, these mechanisms generalize to supervised fine-tuned models. We hope our work contributes to the development of more transparent and reliable LLMs.

---

[3]Note that these may not be the optimal hyperparameters. In fact, extending steering from layers 6-14 to 6-20 reduces retraction recall on Belief+ CELEBRITY set from 0.5763 to 0.2300.

ETHICS STATEMENT

This work does not raise specific ethical concerns. All datasets used in this study (WIKIDATA, CELEBRITY, and UTQA) are publicly available and do not contain private or sensitive information. Our analysis focuses on model behavior and does not involve human subjects.

REPRODUCIBILITY STATEMENT

This work is fully reproducible. We provide the source code in the supplementary material and include detailed descriptions of data construction (Appendix B.1 B.2), retraction evaluation (Appendix B.6), probing (Section 4.1), steering (Appendix B.3), patching (Appendix B.4), and supervised fine-tuning (Appendix B.5) in this paper.

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

## A  ADDITIONAL STATEMENTS

### A.1  THE USE OF LLMS

This paper only used LLMs to polish writing. All original content came from the authors themselves.

### A.2  LIMITATIONS

There are several limitations for future research. First, although different LLMs share the same overall retraction mechanism—being causally influenced by the model's internal belief—the specific layers where this influence is most pronounced vary across models. As shown in Appendix B.3, steering at early to mid layers is effective for Llama3.1-8B and Qwen2.5-7B, whereas Olmo2-7B requires intervention at higher layers to elicit stronger retraction. These differences likely stem from variations in training recipes, including data and optimization strategies used.

Second, our analysis focuses on short-answer knowledge-related question answering tasks. One natural extension is to long-form generation, such as "Name 15 politicians who were born in New York City". This introduces new challenges, including how to accurately locate each generated answer and how to isolate the influence of earlier outputs on later ones. Moreover, as many self-correction studies target reasoning tasks, it would be valuable to examine whether our findings generalize to that domain. However, caution is needed to disentangle limitations in retraction from other capabilities required for reasoning tasks, such as arithmetic computation and problem understanding.

## B  EXPERIMENTAL DETAILS

### B.1  DETAILS OF ORIGINAL DATASETS

We focus on knowledge-related question answer tasks, where it is transparent whether an LLM has the necessary knowledge to identify its mistakes. To facilitate the study of retraction, we collect questions from WIKIDATA and CELEBRITY, which are easy to induce hallucinations. The number of questions in each split of the datasets is reported in Table 9

|  | # Train | # Test |
|---|---|---|
| WIKIDATA | 2000 | 1160 |
| CELEBRITY | 1584 | 800 |

Table 9: Number of questions.

**WIKIDATA.** WIKIDATA was originally proposed by Dhuliawala et al. (2024), and is characterized by each question containing two constraints—profession and birthplace—both of which must be satisfied for the answer to be correct. This makes the task challenging for LLMs, resulting in relatively low accuracy (Yüksekgönül et al., 2024). However, the original dataset was not publicly released. To reconstruct it, we collect a set of popular professions and cities, and generate new questions by pairing them. We retain only those combinations for which a correct exists. For accuracy evaluation, we query the Wikidata API[4]. An example question is:

    Name a writer who was born in Oran, Algeria.

**CELEBRITY.** CELEBRITY was originally introduced by Berglund et al. (2024). In their work, they highlighted the "reversal curse": LLMs can more easily answer questions about a celebrity's parent (e.g., "Who is Tom Cruise's mother?") than the reverse (e.g., "Who is Mary Lee Pfeiffer's son?", where the correct answer is Tom Cruise). We focus on the reverse questions. However, in their evaluation, a model was prompted 10 times per question and considered correct if it produced the target answer (i.e., the celebrity child) at least once. This evaluation cannot determine whether an answer is correct. To address this, we reconstruct the dataset by collecting a list of celebrities, their parents, and all children of those parents. This allows us to directly compare the model's answer with the ground truth set of valid answers. An example question is:

    Name a child of Joe Jackson.

---

[4]https://query.Wikidata.org/

## B.2 DETAILS OF CONTINUATION DATASETS

In addition to constructing wrong examples, we also include correct examples with factually correct answers, to evaluate over-retraction and support in-distribution SFT in Section 6. To avoid bias during SFT, we aim to balance the number of correct and wrong examples. Because model-generated answers are often incorrect on these two datasets, we supplement the correct examples by selecting gold answers for which the model answers the corresponding verification questions correctly.

We build the training and test sets using questions from the train and test splits of the original datasets, respectively. However, Qwen2.5-7B and Olmo2-7B know little about the correct answers in CELEBRITY. Consequently, as shown in Table 1, their CELEBRITY test sets are imbalanced, containing 1,000 incorrect examples and a smaller number of correct ones. Additionally, these two models lack a CELEBRITY training set, which only impacts the in-distribution SFT experiments in Section 6. Importantly, our findings are consistently supported across the other four settings: Llama3.1-8B + WIKIDATA, Llama3.1-8B + CELEBRITY, Qwen2.5-7B + WIKIDATA, and Olmo2-7B + WIKIDATA.

## B.3 HYPERPARAMETERS FOR STEERING

The choice of steering layers and strength is critical to clearly demonstrate the effect of steering without compromising the model's original capabilities. Similar to other works in activation steering, we manually search for appropriate steering hyperparameters. Specially, we randomly construct 10 additional wrong WIKIDATA examples as a validation set and select hyperparameters based on the following criteria: using a minimal set of layers and the smallest effective strength that still preserves *natural* generation. Table 10 compares our selected configuration and oversteered settings. Although hyperparameters are chosen using only wrong WIKIDATA examples for negative belief steering, they generalize well to positive belief steering, positive examples, the CELEBRITY dataset, and the *is*-appended setting, demonstrating the generalizability of the belief steering. While more exhaustive hyperparameter sweeps could potentially identify even better settings, we find that the current choices produce stable and consistent effects across different datasets. The final choices are listed in Table 11.

| Question | Properly Steered Response | Oversteered Response |
|---|---|---|
| Name a poet who was born in Panama City, Panama. | Giannina Braschi is not the answer, however, Giannina Braschi was born in San Juan, Puerto Rico. | Giannina Braschi **nor** Omar Cabezas are not the answer I am looking for. |
| Name a television actor who was born in Johannesburg, South Africa. | Sterling K. Brown isn't from Johannesburg, South Africa. The actor born there is Sharlto Copley. | Sterling K. Brown **Nope**, that's incorrect. Let me try again. Jonny Lee Miller was born in Johannesburg, South Africa. |

Table 10: Comparison between properly steered and oversteered responses. (1) When steering Llama3.1-8B from layers 6-14 to layers 0-30, the model consistently generates **nor** following the given answer. Although it can be regarded as a retraction, the phrasing is unnatural. (2) When increasing the steering strength $\alpha$ from 1.5 to 3.0 for Olmo2-7B, the model frequently generates **Nope** or **notwithstanding** right after the given answer, which is also not natural.

| Model | Layers | Strength $\alpha$ |
|---|---|---|
| Llama3.1-8B | 6-14 | 1.2 |
| Qwen2.5-7B | 10-18 | 2.5 |
| Olmo2-7B | 8-30 | 1.5 |

Table 11: Steering hyperparameters.

### B.4 Implementation Details for Patching

**Patching Attention Weights.** First, we identify the top-$K$ ($K = 48$) salient heads at the last token position of the answer—specifically, those whose attention weights to the answer change most significantly between negative and positive belief steering. Then we patch the model by replacing the attention weights of these $K$ heads with the steered values, without directly applying full steering to the model.

**Patching Attention Value Vectors.** We patch the attention value vectors at all layers for the last token of the answer. Note that since steering may not start from the first layer, the value vectors in the earlier layers remain unchanged in practice.

### B.5 Implementation Details for Supervised Fine-Tuning

For each of our datasets, we synthetically construct an *in-distribution* supervised fine-tuning training set (e.g., training on Wikidata training set and evaluating on Wikidata test set). Specifically, we append " is the correct answer." to correct examples that contain factually correct answers in the training dataset, and " is not the correct answer." to wrong examples.

We fine-tune models using LoRA (Hu et al., 2022) for 2 epochs with a learning rate of $1e-4$ and a batch size of 8, implemented via LLaMA-Factory (Zheng et al., 2024). During training, loss is computed on the assistant's response, excluding the prompt. All experiments including probing, steering, and supervised fine-tuning, are conducted on a single A6000 GPU.

The results are shown in Table 8 and 19. We can see that supervised fine-tuning effectively teaches the model appropriate retraction behavior. The model learns to distinguish between factually correct and incorrect answers and respond accordingly, i.e., saying "is the correct answer" to correct answers and saying "is not the correct answer" to incorrect ones.

## B.6  RETRACTION DETECTION USING LLM-AS-A-JUDGE

The prompt for retraction detection using Llama3.3-70B-Instruct is shown below. Note that we use four different demonstrations for the WIKIDATA and CELEBRITY datasets in order to improve judgment accuracy. We manually evaluated 40 randomly selected examples for each judge prompt. Llama3.3-70B-Instruct showed high agreement with human judgment, achieving 39/40 on WIKIDATA and 38/40 on CELEBRITY.

---

**Retraction Detection Prompt**

[t] You will be given a user's question, an assistant's response, and the initial answer provided in the assistant's response. Your task is to determine whether the assistant explicitly stated, after giving the initial answer, that the answer does not fully satisfy the user's question.

**Guidelines:**

1. Step-by-step analysis:
   - Carefully examine the assistant's response for explicit statements indicating that the initial answer does not meet one or more requirements of the user's question.
   - Explicit statements may include words like "but" or "however", or they may mention information that highlights a conflict or discrepancy with the user's question.

2. Focus only on the assistant's statements. Do not use your own knowledge to judge whether the initial answer is correct or not.

**Output Format:**

1. **Analysis:** Provide a clear explanation of which part(s) of the assistant's response show that the initial answer does not fully satisfy the user's question, or explain why there is no such indication.

2. **Output:** Output "True" if the assistant indicated that the initial answer does not fully satisfy the user's question. Otherwise, output "False".

Here are a few examples for reference:

**Example 1:**
User Question: Name an actor who was born in Sheffield, United Kingdom.
Assistant Response: Michael Palin.
Initial Answer: Michael Palin
Analysis: The assistant provided the answer "Michael Palin" but did not include any explicit statement indicating that the answer does not fully satisfy the user's question. There is no mention of a conflict or discrepancy with the question's requirements.
Output: False

**Example 2:**
User Question: Name an association football manager who was born in Sydney, Australia.
Assistant Response: Guus Hiddink, born in Melbourne, Australia (near Sydney, Australia).
Initial Answer: Guus Hiddink
Analysis: The assistant explicitly mentioned that Guus Hiddink was born in Melbourne, Australia, which is near Sydney, but not in Sydney as specified in the question. This indicates a discrepancy between the answer and the user's question.
Output: True

*[Omit the other two examples here...]*

Now, process the following:
User Question: {question}
Assistant Response: {response}
Initial Answer: {model's answer}

---

## C ADDITIONAL RESULTS

### C.1 PROBING PLOTS

Since the retraction recall of Qwen2.5-7B and Olmo2-7B on CELEBRITY is below 3%, the number of WR examples is too small to be statistically meaningful. Therefore, we report AUROC of belief scores for these two models only on WIKIDATA, as shown in Figure 5. Both models consistently present high correlation between belief scores and retraction.

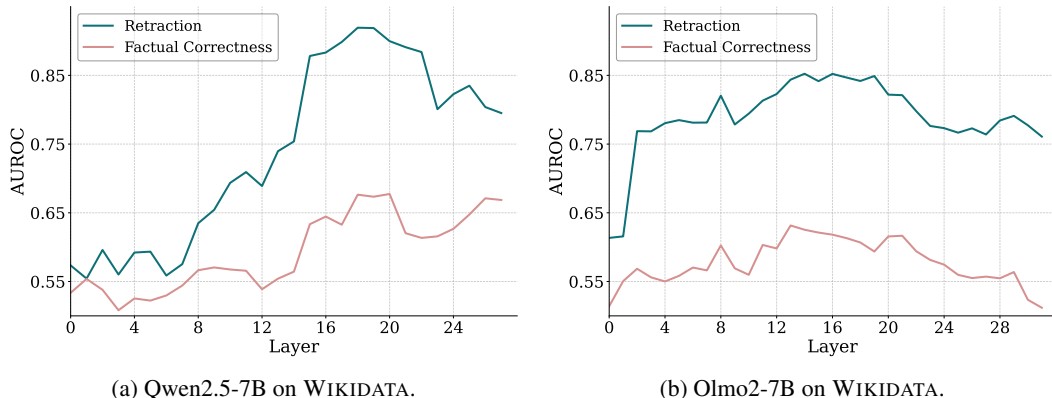

(a) Qwen2.5-7B on WIKIDATA.    (b) Olmo2-7B on WIKIDATA.

Figure 5: Layer-wise AUROC of belief scores for factual correctness and retraction of Qwen2.5-7B and Olmo2-7B on the WIKIDATA test set.

### C.1.1 ROBUSTNESS TO MODEL SCALE

To evaluate whetehr our findings generalize to larger LLMs, we additionaly study Llama3.1-70B-Instruct. We construct its WIKIDATA continuation test set of 4,492 examples with balanced correct and wrong answers, and test its retraction behavior. The model achieves a **retraction precision of 0.9126 and recall of 0.5303**, outperforming the smaller Llama3.1-8B-Instruct (precision 0.9012, recall 0.2579), yet still leaves substantial room for improvement.

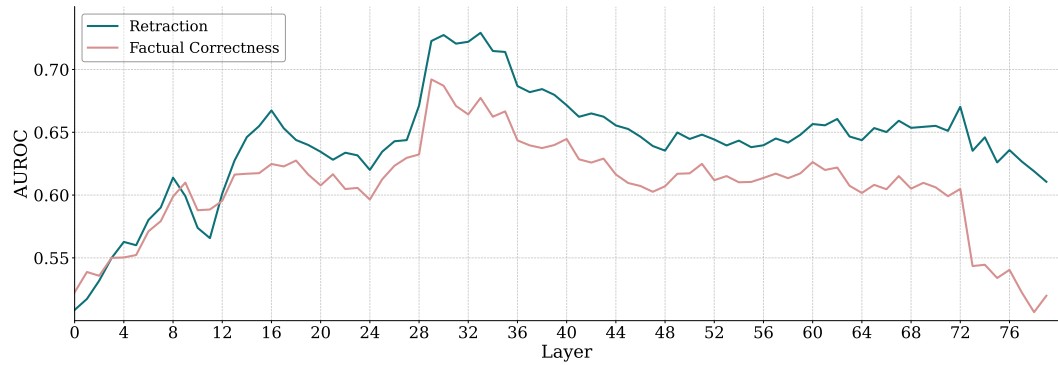

Figure 6: Layer-wise AUROC of belief scores for factual correctness and retraction of Llama3.1-70B-Instruct on WIKIDATA.

We next examine whether belief continues to explain retraction behavior at this scale through the probing experiment. Given that Llama3.1-70B-Instruct contains 80 layers with 8192-dimensional hidden states (vs. 32 layers and 4096 dimensions in Llama3.1-8B-Instruct), to mitigate overfitting, we expand the UTQA training set from 2,400 to 8,000 examples by additionally incorporating samples from ANLI (Nie et al., 2020), AG News (Zhang et al., 2015), MRPC (Wang et al., 2019), SQuAD (Rajpurkar et al., 2016), OpenBookQA (Mihaylov et al., 2018), Winogrande (Sakaguchi

et al., 2020), and Truthful QA (Lin et al., 2022). We train an independent *single-linear* probe at each layer, and the resulting AUROC scores are shown in Figure 6.

We can see that the belief probes achieve moderately strong performance on predicting retraction behavior in layers 29-35, and consistently outperform factual-correctness prediction, mirroring patterns observed in the 8B variant and in other model families. Their predictive power is somewhat lower than probes on smaller models, possibly because larger models are more expressive and encode multiple entangled features at the coarse layer level. More fine-grained extraction of belief representations, such as at the head level or using sparse autoencoders (Huben et al., 2024; Shu et al., 2025), may reveal more generalizable belief signals.

Overall, these results indicate that **the retraction mechanism and its connection to belief remain consistent from smaller to larger LLMs across families.**

## C.2 UNCERTAINTY VS. RETRACTION

An intuitive hypothesis is that retraction may also be related to the model's uncertainty. In this section, we examine this relationship and report the AUROC of uncertainty scores against retraction labels. All experiments are conducted using LLaMA3.1-8B on the WIKIDATA dataset.

**Token-Level Entropy.** We first examine whether higher uncertainty in an answer, as measured by token-level entropy, is associated with a greater likelihood of retraction. For each answer in the continuation dataset, we compute the average token-level entropy as follows:

$$\text{Entropy} = -\frac{1}{T}\sum_{t=1}^{T}\sum_{v\in V} p_t(v)\log p_t(v),$$

where $T$ is the number of tokens in the answer, $V$ is the vocabulary, and $p_t(v)$ is the model's predicted probability of token $v$ at position $t$.

Using token-level entropy to predict retraction yields an AUROC of **0.518**, only marginally above random chance (AUROC $= 0.5$). This result indicates no meaningful correlation with retraction.

**Consistency-Based Uncertainty.** Next, we assess consistency-based uncertainty (Xiong et al., 2024) by sampling $n = 5$ answers per question and defining an answer's uncertainty as:

$$\text{Uncertainty}(a_i) = 1 - \frac{|a_i|}{n},$$

where $|a_i|$ is the number of times the same answer appears among the five samples.

Predicting retraction using consistency-based uncertainty yields an AUROC of **0.533**, reflecting weak discriminative capacity.

**Inter-Answer Entropy.** Additionally, we investigate whether uncertainty can identify questions where the model is more likely to exhibit retraction behavior. We measure question-level uncertainty by computing the entropy over the five generated answers per question, following (Kuhn et al., 2023; Xiong et al., 2024):

$$\text{Entropy}(q) = -\sum_{a\in A_q} p(a)\log p(a),$$

where $A_q$ is the set of unique answers generated for question $q$, and $p(a)$ is the relative frequency of answer $a$ among the five samples. To handle semantic equivalence, we extract answers (e.g., person names in the WIKIDATA dataset) from model responses using Llama-3.3-70B-Instruct, rather than relying on an NLI model as in Kuhn et al. (2023).

Inter-answer entropy achieves an AUROC of **0.505** for predicting retraction, offering little predictive values.

Compared to all the methods discussed above, our belief probe scores show a significantly higher correlation with retraction behavior. Uncertainty, by contrast, may require more precise definitions and further study to uncover its potential connection to retraction.

## C.3 ROBUSTNESS TO PROMPTING AND DECODING VARIATION

To assess the robustness of belief-retraction dynamics, we evaluate how prompt templates and decoding hyperparameters affect both the model's baseline retraction behavior and the effectiveness of belief steering. All experiments in this section use Llama3.1-8B-Instruct and WIKIDATA.

### C.3.1 ROBUSTNESS TO PROMPT VARIATION

We test two standard prompt phrasings along with an adversarial variant designed to introduce an incorrect statement before the question to potentially bias the model's belief:

- **Original Prompt**: Name an association football player who was born in Naples, Italy.
- **Prompt 1**: Can you name an association football player who was born in Naples, Italy?
- **Prompt 2**: Which association football player was born in Naples, Italy? Just name one.
- **Adv Prompt**: Barack Obama is a politician who was born in New York City, United States. Name an association football player who was born in Naples, Italy.

| | Prompt 1 | | Prompt 2 | | Adv Prompt | |
|---|---|---|---|---|---|---|
| | Prec. | Rec. | Prec. | Rec. | Prec. | Rec. |
| No Steering | 0.8994 | 0.4609 | 0.9211 | 0.0582 | 0.8138 | 0.4725 |
| Belief- Steering | 0.5296 | 0.8935 | 0.5115 | 0.8902 | 0.5280 | 0.8319 |
| Belief+ Steering | 0.9412 | 0.0266 | 1.0000 | 0.0017 | 0.9091 | 0.0166 |

Table 12: Steering results under different prompt variants.

As shown in Table 12, across all templates, belief- steering greatly leads to more retraction, while belief+ steering suppresses retraction. This demonstrates that **belief steering is robust to changes in prompt phrasing.**

### C.3.2 ROBUSTNESS TO DECODING VARIATION

We further analyze the effect of decoding hyperparameters. In addtion to greedy decoding, we evaluate temperature sampling (temperature $= 0.7$, top-p $= 0.95$), repeating each run three times across different seeds.

| | Precision | Recall |
|---|---|---|
| Greedy Decoding | 0.9012 | 0.2579 |
| Temperature Sampling | $0.8814_{(0.0105)}$ | $0.3128_{(0.0088)}$ |
| Temperature Sampling with Belief– Steering | $0.5257_{(0.0013)}$ | $0.8980_{(0.0101)}$ |
| Temperature Sampling with Belief+ Steering | $0.9551_{(0.0024)}$ | $0.0355_{(0.0019)}$ |

Table 13: Steering results under different decoding hyperparameters. Subscripts indicate standard deviation.

Table 13 suggests that **the link between belief and retraction holds consistently across decoding hyperparameters.**

## C.4 OTHER STEERING DIRECTIONS

Except for the belief direction, we also try another two directions that are likely to affect retraction behavior. (1) **WIKIDATA retraction direction**: The positive examples are those that the model actually retracts from the WIKIDATA training set, and negative examples are those that the model does not retract. (2) **WIKIDATA correctness direction**: The positive examples contain factually correct answers from the WIKIDATA training set, and negative examples contain factually incorrect answers. We search for the best hyperparameters as described in Appendix B.3, and find that those used in belief steering yield the best retraction performance among the hyperparameters we explored for Llama3.1-8B. We show the results in Table 14.

| | WIKIDATA | | CELEBRITY | |
|---|---|---|---|---|
| | Precision | Recall | Precision | Recall |
| No Steering | 0.9012 | 0.2579 | 0.7722 | 0.1477 |
| Belief- | 0.5157 | 0.9268 | 0.4803 | 0.7676 |
| WIKIDATA Retraction+ | 0.5029 | 0.7321 | 0.5638 | 0.6634 |
| WIKIDATA Correctness- | 0.5075 | 0.7903 | 0.5707 | 0.5569 |
| Belief+ | 1.0000 | 0.0067 | 0.5217 | 0.0291 |
| WIKIDATA Retraction- | 0 | 0 | 0.6667 | 0.0048 |
| WIKIDATA Correctness+ | 0.5000 | 0.0083 | 0.6667 | 0.0097 |

Table 14: Retraction Performance for Llama3.1-8B on continuation test sets.

| | WIKIDATA | | CELEBRITY | |
|---|---|---|---|---|
| | Prec. | Rec. | Prec. | Rec. |
| No Steer | 0.8824 | 0.1119 | 0.9667 | 0.0290 |
| Belief- | 0.5051 | 0.8358 | 0.8547 | 0.7000 |
| Belief+ | 1.0000 | 0.0131 | 1.0000 | 0.0090 |

| | WIKIDATA | | CELEBRITY | |
|---|---|---|---|---|
| | Prec. | Rec. | Prec. | Rec. |
| No Steer | 0.9881 | 0.1317 | 0.8824 | 0.0150 |
| Belief- | 0.5206 | 0.7619 | 0.8217 | 0.7420 |
| Belief+ | 1.0000 | 0.0016 | 0 | 0 |

Table 15: Retraction Performance for Qwen2.5-7B on continuation test sets.

Table 16: Retraction Performance for Olmo2-7B on continuation test sets.

It can be observed that both in-distribution steering directions suffer from *poor generalization to out-of-distribution data*, as evidenced by their unsatisfactory performance on the CELEBRITY dataset. Additionally, for the WIKIDATA retraction direction, the mean hidden state representations may be unrepresentative due to (1) a limited number of retracted examples serving as positive examples, and (2) the use of in-distribution data. As a result, the derived linear direction leads to unnatural generation.

Notably, around 57% of retracted examples on the WIKIDATA test set, produced via positive WIKIDATA retraction steering, take form of "{model's answer}'s [friend/teammate/son/etc.]". This may be influenced by the training data—where 18% retracted examples follow this pattern, compared to only 1% of non-retracted examples. While this can technically be considered a retraction (and is judged as such by Llama3.3-70B-Instruct), the phrasing is awkward. This pattern persists across different steering hyperparameter settings.

## C.5 PATCHING RESULTS

Patching results under *is*-appended setting for Qwen2.5-7B and Olmo2-7B are shown in Table 17 and 18. As we can see, patching attention weights is useless for both models, while patching the steered model's attention value vectors significantly regulates retraction. Note that for Olmo2-7B, we increase the original $\alpha$ from 1.5 to 5.0 to make belief steering effective under *is*-appended setting. This implies that, at $\alpha = 1.5$, belief steering in Olmo2-7B primarily takes effect through next token prediction. Nevertheless, larger $\alpha$ values still modify the attention value vectors in a manner consistent with our overall conclusions. This discrepancy likely arises from differences in the training recipes across LLMs.

| | WIKIDATA | | CELEBRITY | | | WIKIDATA | | CELEBRITY | |
|---|---|---|---|---|---|---|---|---|---|
| | Prec. | Rec. | Prec. | Rec. | | Prec. | Rec. | Prec. | Rec. |
| No Steer | 0.8500 | 0.0951 | 1.0000 | 0.0320 | No Steer | 1.0000 | 0.0730 | 1.0000 | 0.0130 |
| *belief-* | | | | | *belief-* | | | | |
| Patch W | 0.8846 | 0.0877 | 1.0000 | 0.0340 | Patch W | 0.9767 | 0.0667 | 1.0000 | 0.0140 |
| Patch V | 0.5209 | 0.9049 | 0.8371 | 0.3700 | Patch V | 0.5012 | 0.9762 | 0.8230 | 0.9580 |
| Full Steer | 0.5079 | 0.8955 | 0.8601 | 0.7560 | Full Steer | 0.5140 | 0.5810 | 0.6980 | 0.1410 |
| *belief+* | | | | | *belief+* | | | | |
| Patch W | 0.8814 | 0.0970 | 1.0000 | 0.0310 | Patch W | 1.0000 | 0.0619 | 1.0000 | 0.0170 |
| Patch V | 0.9375 | 0.0280 | 1.0000 | 0.0270 | Patch V | 0.9200 | 0.0365 | 0.9545 | 0.0210 |
| Full Steer | 0.9444 | 0.0317 | 1.0000 | 0.0210 | Full Steer | 1.0000 | 0.0048 | 1.0000 | 0.0150 |

Table 17: Patching results for Qwen2.5-7B under the *is*-appended setting.

Table 18: Patching results for Olmo2-7B under the *is*-appended setting with $\alpha = 5.0$.

## C.6 SUPERVISED FINE-TUNING RESULTS

### C.6.1 SFT RESULTS FOR QWEN AND OLMO

Building on Llama3.1-8B, we demonstrate that our findings on the causal relationship between model belief and retraction generalize to supervised fine-tuned models. This is further supported by results from Qwen2.5-7B and Olmo2-7B. As shown in Table 19, the same belief steering directions remain effective after fine-tuning. Additionally, Figure 7 indicates that supervised fine-tuning leads to more accurate internal beliefs.

| | Qwen2.5-7B | | Olmo2-7B | |
|---|---|---|---|---|
| | Precision | Recall | Precision | Recall |
| Baseline | 0.8824 | 0.1119 | 0.9881 | 0.1317 |
| SFT | 0.8350 | 0.7929 | 0.8869 | 0.8460 |
| Belief- for SFT | 0.5023 | 1.0000 | 0.5179 | 0.9873 |
| Belief+ for SFT | 0.9391 | 0.2015 | 0.9934 | 0.2381 |

Table 19: In-distribution supervised fine-tuning results for Qwen2.5-7B and Olmo2-7B on WIKIDATA.

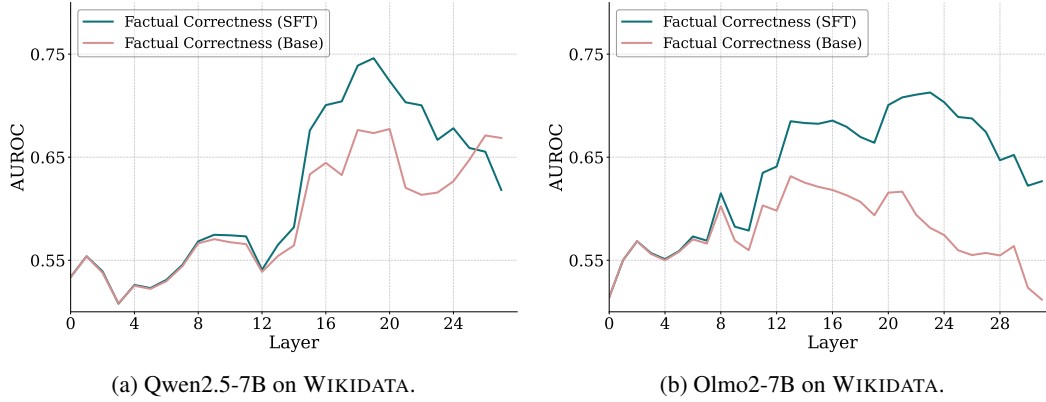

(a) Qwen2.5-7B on WIKIDATA.   (b) Olmo2-7B on WIKIDATA.

Figure 7: Layer-wise AUROC of belief scores for fatucal correctness of Qwen2.5-7B and Olmo2-7B (Base), and their fine-tuned variants (SFT).

### C.6.2 PRACTICAL APPLICATION

The continuation setting is a synthetic setup designed to facilitate controlled study. Here, we consider a more realistic scenario: given a question, what does SFT achieve? As shown in Table 20, while SFT does not improve accuracy as no new knowledge is introduced, it substantially enhances retraction performance, thereby making the model more reliable.

| | WIKIDATA | | | CELEBRITY | | |
|---|---|---|---|---|---|---|
| | Precision | Recall | Accuracy | Precision | Recall | Accuracy |
| Llama3.1-8B | 0.9928 | 0.2715 | 0.0841 | 0.8125 | 0.0884 | 0.3163 |
| Llama3.1-8B SFT | 0.9481 | 0.7079 | 0.0840 | 0.9774 | 0.8162 | 0.2261 |

Table 20: SFT results for Llama3.1-8B in a realistic setting.

### C.7 GENERALIZATION TO MATH REASONING

Although our analysis primarily focuses on factoid QA, we further investigate whether belief steering extends to math reasoning, using the GSM8k dataset (Cobbe et al., 2021).

| | Acc |
|---|---|
| Belief- Steering | 37.77% |
| Belief+ Steering | 17.55% |

Table 21: Steering results on GSM8k.

We first collect Llama3.1-8B-Instruct's trajectories that produce *incorrect* final answer via greedy decoding, and use GPT-4.1 to annotate the first incorrect token. This results in a total of 188 examples (where accuracy = 0 with no steering by construction). We then apply negative belief steering, without altering any other generation setting, and evaluate accuracy on this subset. To moderately amplify the effect, we steer layers 0-18 and intervene on the first incorrect token plus its preceding nine tokens (a randomly selected hyperparameter). Since math reasoning depends on multi-step computation, modifying early hidden states can propagate and influence later inference. We also report positive belief steering to reveal unintended effects.

---

**Case Study**

**[QUESTION]**
Carla is downloading a 200 GB file. Normally she can download 2 GB/minute, but **40% of the way through the download, Windows forces a restart** to install updates, which takes 20 minutes. Then Carla has to restart the download from the beginning. How long does it take to download the file?

**[RESPONSE] – No Steering ✗**
... 3. After the restart, Carla has to download the remaining 60% of the file (100% - 40% = 60%)...

**[RESPONSE] – Belief- Steering ✓**
... 3. After the restart, Carla has to download the remaining 60%[a] of the file, but Carla has to restart the download from the beginning. So Carla has to download the entire 200 GB again...

**[RESPONSE] – Belief+ Steering ✗**
... 3. After the restart, Carla has to download the remaining 60% .
   60% of 200 GB is 0.6 * 200 = 120 GB...

---
[a]Steering is applied at this token and the preceding nine tokens.

---

Recall that the belief vector is derived from a quite different dataset UTQA. Negative belief steering still activates retraction behavior in math reasoning and improve the final accuracy by around 20%, **demonstrating the robustness and generalization of our interpretability findings.**

