# OpenReview forum: "When Do LLMs Admit Their Mistakes? Understanding the Role of Model Belief in Retraction"
_ICLR.cc/2026/Conference — ICLR 2026 Conference Withdrawn Submission_

### Official Review · Reviewer_VBRM · 2025-10-21

**Soundness:** 4
**Presentation:** 4
**Contribution:** 4
**Rating:** 6
**Confidence:** 4

**Summary:**

This paper studies when and why large language models (LLMs) retract their mistakes—defined as spontaneous acknowledgments of incorrect answers. The authors construct model-specific continuation datasets from WIKIDATA and CELEBRITY benchmarks, where models generate potentially wrong answers and are then observed to see if they retract them without external prompting. Overall, the paper provides a principled, mechanistic understanding of why models sometimes refuse to retract even when they “know better.”

**Strengths:**

- Positions “retraction” as a measurable, meaningful behavioral metric for model reliability.

- Uses steering and patching methods to demonstrate directional control over behavior.

- Provides concrete evidence that attention value vectors, not just weights, mediate belief propagation.

- Evaluates multiple model families, increasing robustness.

- Links findings to SFT improvements, suggesting paths for aligning model introspection with truthfulness.

- Extensive appendices, code availability, and transparent methodology.

**Weaknesses:**

- Focuses on factual QA; results may differ for open-ended or reasoning-heavy tasks (e.g., math or multi-step reasoning).

- While linear probes capture useful signals, belief may conflate confidence, calibration, and factual recall.

- Although consistent, evaluation could benefit from human verification for robustness.

- Steering effects may differ across architectures or prompt styles.

- Retracting only ~25% of wrong answers initially limits downstream applicability, even if mechanisms are well-understood.

**Questions:**

- How robust are belief-based probes to changes in prompt style or model sampling temperature?

- Could non-linear or attention-based probes (e.g., small MLPs) capture belief signals more accurately?

- How does belief steering generalize to reasoning or creative tasks, where “wrong” answers are less well-defined?

- Does the causal influence of belief persist in RLHF-tuned models, which are already calibrated for truthfulness?

- Have you examined whether cross-model belief vectors (e.g., between Qwen and Llama) share a universal direction?

**Details Of Ethics Concerns:**

The datasets (WIKIDATA, CELEBRITY, UTQA) are public and non-sensitive. The study focuses purely on model behavior and interpretability. There are no human subjects or privacy issues.

---

> ### Author Response · Authors · 2025-11-21
> **[1/3] Response to Reviewer VBRM**
>
> Thank you for your insightful comments. Let us try to address your concerns below.
>
> ### Question 1 and Weakness 4: Prompt and Decoding Sensitivity
>
> To examine how prompt templates and decoding hyperparameters influence the model’s original retraction performance, and whether belief still causally affects retraction, we conduct experiments using various prompt formulations and decoding settings. In brief, although both prompt templates and decoding hyperparameters do affect the original retraction performance, the model still does not retract enough, and belief steering works in all scenarios. All experiments below use Llama3.1-8B-Instruct and Wikidata.
>
> **1. Prompt Sensitivity**
>
> We test three prompt templates:
>
> **original_prompt:** Name an association football player who was born in Naples, Italy.
>
> **new_prompt1:** Can you name an association football player who was born in Naples, Italy?
>
> **new_prompt2:** Which association football player was born in Naples, Italy? Just name one.
>
> **adv_prompt:** Barack Obama is a politician who was born in New York City, United States. Name an association football player who was born in Naples, Italy. (Adversarial prompt template: it adds an incorrect statement before the question to potentially bias the model’s belief.)
>
> Results are shown below:
>
> | new_prompt1 | Retraction Precision | Retraction Recall |
> | --- | --- | --- |
> | No Steering | 0.8994 | 0.4609 |
> | Belief− Steering | 0.5296 | 0.8935 |
> | Belief+ Steering | 0.9412 | 0.0266 |
>
> | new_prompt2 | Retraction Precision | Retraction Recall |
> | --- | --- | --- |
> | No Steering | 0.9211 | 0.0582 |
> | Belief− Steering | 0.5115 | 0.8902 |
> | Belief+ Steering | 1.0 | 0.0017 |
>
> | adv_prompt | Retraction Precision | Retraction Recall |
> | --- | --- | --- |
> | No Steering | 0.8138 | 0.4725 |
> | Belief− Steering | 0.5280 | 0.8319 |
> | Belief+ Steering | 0.9091 | 0.0166 |
>
> Consistent with the observations in Section 4.2, negative belief steering (i.e., belief- steering) leads the model to retract significantly more often, while positive belief steering (i.e., belief+ steering) causes the model to almost never retract. This demonstrates that **belief steering is robust to changes in prompt phrasing.**
>
> **2. Decoding Sensitivity**
>
> In this paper, we use greedy decoding for determinism. Here, we supplement the analysis with temperature sampling (temperature=0.7, top-p=0.95), repeated three times with different random seeds. Numbers in parentheses indicate the standard deviation.
>
> |  | Retraction Precision | Retraction Recall |
> | --- | --- | --- |
> | Greedy Decoding | 0.9012 | 0.2579 |
> | Temperature Sampling | 0.8814 (0.0105) | 0.3128 (0.0088) |
> | Temperature Sampling with Belief- Steering | 0.5257 (0.0013) | 0.8980 (0.0101) |
> | Temperature Sampling with Belief+ Steering | 0.9551 (0.0024) | 0.0355 (0.0019) |
>
> These results show that **the link between belief and retraction holds consistently across decoding hyperparameters.**
>
> ### Question 2: More Complex Probes
>
> Thank you for the question. We intentionally adopt the simplest probe architecture, a single linear layer, to reveal features genuinely used by the generation model. While non-linear and more complex probes may perform better in in-distribution classification tasks, they (1) tend to overfit and generalize poorly to out-of-distribution data, and more importantly, (2) “bears the risk that the classifier infers features that are not actually used by the network” [1,2]. We therefore choose linear probes, which are well-suited for interpretability-focused analysis.
>
> [1] Probing Classifiers: Promises, Shortcomings, and Advances.
>
> [2] Visualisation and'diagnostic classifiers' reveal how recurrent and recursive neural networks process hierarchical structure.

---

> ### Author Response · Authors · 2025-11-21
> **[2/3] Response to Reviewer VBRM**
>
> ### Question 3 and Weakness 1: Reasoning Tasks
>
> Motivated by your comment, we apply belief steering to GSM8k, a math-reasoning dataset, to explore how belief affects the model’s reasoning.
>
> We first collect Llama3.1-8B-Instruct’s trajectories that produce *incorrect* final answer via greedy decoding, and use GPT-4.1 to annotate the first incorrect token. This results in a total of 188 examples (where accuracy = 0 with no steering by construction). We then apply negative belief steering, without altering any other generation setting, and evaluate accuracy on this subset. To moderately amplify the effect, we steer layers 0-18 and intervene on the first incorrect token plus its preceding nine tokens (a randomly selected hyperparameter). Since math reasoning depends on multi-step computation, modifying early hidden states can propagate and correct later inference. We also report positive belief steering to reveal unintended effects.
>
> **Results:**
>
> |  | Acc |
> | --- | --- |
> | Belief- Steering | 37.77% |
> | Belief+ Steering | 17.55% |
>
> **Case Study:**
>
> > Question: Carla is downloading a 200 GB file. Normally she can download 2 GB/minute, but **40% of the way through the download, Windows forces a restart** to install updates, which takes 20 minutes. Then Carla has to restart the download from the beginning. How long does it take to download the file?
>
> No Steering: wrong
>
> > Assistant: ...3. After the restart, Carla has to download the remaining 60% of the file (100% - 40% = 60%)…
>
> Negative Belief Steering: correct *(steering takes effect on the italicized part)*
>
> > Assistant: ...3. After the restart, Carla has to download the remaining 60% *of the file, but Carla has to restart the download from the beginning. So Carla has to download the entire 200 GB again…*
>
> Positive Belief Steering: wrong *(steering takes effect on the italicized part)*
>
> > Assistant: ...3. After the restart, Carla has to download the remaining 60% *.*
> >
> >    *60% of 200 GB is 0.6 * 200 = 120 GB…*
>
> Recall that the belief vector is derived from a quite different dataset UTQA. Negative belief steering still activates retraction behavior in math reasoning and improve the final accuracy by around 20%, **demonstrating the robustness and generalization of our interpretability findings.**
>
> ### Question 4: RLHF-Tuned Models
>
> Yes, our experiments on models of different families show that the causal influence of belief persists in RLHF-tuned models and models that are calibrated for truthfulness.
>
> For example, Qwen2.5-7B-Instruct [1] performs online RL to improve truthfulness as described below:
>
> > The Online RL phase leverages the reward model’s ability to detect nuances in output quality, including truthfulness, …
>
> Olmo2-7B-Instruct uses DPO on preference data emphasizing truthful responses [2].
>
> Despite their different architectures, training data, and training procedures, all models exhibit the same retraction mechanism: internal belief causally drives external retraction behavior.
>
> [1] Qwen2.5 Technical Report.
>
> [2] 2 OLMo 2 Furious.
>
> ### Question 5: Cross-Model Belief Vectors
>
> Thank you for this thoughtful question. We agree that cross-model steering is an interesting direction. In this work, we focus on within-model causal analysis, because belief vectors depend on model-specific layer geometry and hidden dimensionality, making direct transfer non-trivial without an alignment mapping. For example, Llama3.1-8B-Instruct (4096-d) and Qwen2.5-7B-Instruct (3584-d) differ both in dimensionality and in where belief is represented across layers as shown in Figure 3(a) and Figure 5.
>
> While we did not perform cross-model steering experiments due to these architectural mismatches, we did evaluate the phenomenon across multiple model families (Llama, Qwen, Olmo) and found that each model independently exhibits the same underlying retraction mechanism.

---

> ### Author Response · Authors · 2025-11-21
> **[3/3] Response to Reviewer VBRM**
>
> ### Weakness 3: LLM Judge Reliability
>
> As described in Appendix B.6, we have manually evaluated 40 randomly selected examples for each judge prompt. Llama3.3-70B-Instruct shows high agreement with human judgment, achieving 39/40 on Wikidata and 38/40 on Celebrity.
>
> To further validate judge reliability, we use GPT-4.1-mini to evaluate Llama3.1-8B-Instruct’s retraction performance under no-steering, positive-steering, and negative-steering settings (1202 * 3 = 3606 examples on Wikidata; 826 * 3 = 2478 examples on Celebirty). We then compute agreement between GPT-4.1-mini and Llama3.3-70B-Instruct.
>
> |  | Consistency | Cohen Kappa |
> | --- | --- | --- |
> | Wikidata | 0.9750 | 0.9445 |
> | Celebrity | 0.9730 | 0.9364 |
>
> Llama3.3-70B-Instruct achieves very high agreement with both human annotators and another LLM judge, while being convenient and cost-free.
>
> ---
> Thank you again for your thoughtful and constructive feedback. Our work aims to systematically evaluate LLMs’ retraction behavior and to distinguish retracted from non-retracted wrong answers through the lens of internal representations, thereby making the underlying mechanism more transparent. We establish a causal relationship between the model’s internal belief and its external retraction behavior, which holds for supervised fine-tuning and out-of-distribution math reasoning tasks.
>
> We sincerely appreciate your time and careful review, and we would be glad to address any further concerns you may have.

---

### Official Review · Reviewer_rHvH · 2025-10-26

**Soundness:** 3
**Presentation:** 3
**Contribution:** 2
**Rating:** 4
**Confidence:** 3

**Summary:**

The paper investigates when and why LLMs spontaneously retract their own incorrect answers. For systematic evaluation, the authors propose continuation testbeds by prefilling the answers (e.g., Hillary Clinton [Model generation continues from here...]). The main observation based on this testbed is LLM can retract, but infrequently. By using probing and activation steering, it shows that the LLM’s internal belief reflects when the model internally “believes” it is wrong, it is more likely to retract. Besides, it shows that supervised fine-tuning improves retraction by aligning internal beliefs with factual correctness.

**Strengths:**

1. The paper introduces a new framework for studying LLM reliability as the testbeds for retraction behaviour.

2. This paper proposed a warning for SFT. The connection between belief and retraction holds even after supervised fine-tuning, showing that improved belief calibration enhances factual alignment and transparency.

**Weaknesses:**

1. For table 2, I am curious if the phenomenon is consistent for larger LLM (e.g., meta-llama/Llama-3.1-70B-Instruct).

2. For the training procedure of the linear prob, I think the training dataset (UTQA dataset) is not a continuation dataset. The linear prob trained on this dataset aims to reflect factual correctness but not the retract behaviour, right? The interesting observation is that the linear prob is  highly predictive of whether the model will retract its answer. Any additional and deep explanation for this phenomenon will be very interesting.

3. For activation steering, it shows that changing internal belief alone is enough to change whether the model admits a mistake. I believe some ablation studies should be added to support this claim. For example, we should first split the testbed into two groups by the criterion as if the LLM can do retraction originally, and then study how negative/positive belief steering can change the retraction rate for these two different groups.

4. For supervised fine-tuning, figure 4 shows that the probing of factual correctness can achieve significantly higher accuracies for middle layers (12-24). It is very interesting. Any explanation for this phenomenon?

5. There are several interesting observations in this paper, but most of them are quite facial. Besides, the relation between factual correctness and retraction behaviour is still not explained clearly.

6. The novelty is quite limited. The methods used for analysis, such as linear probing, steering, and patching are not novel.

**Questions:**

Please check the weaknesses.

---

> ### Author Response · Authors · 2025-11-21
> **[1/2] Response to Reviewer rHvH**
>
> Thank you for your detailed and thoughtful comments. Here are responses to your concerns.
>
> ### Weakness 1: Larger LLMs
>
> Motivated by your comment, we additionally evaluate Llama3.1-70B-Instruct to examine whether larger LLM has the ability to retract, yet are still reluctant to retract incorrect generated answers despite having the requisite knowledge (findings from Table 2). Specifically, we first construct its Wikidata continuation test set, comprising 4492 examples: half with factually correct answers and half with factually incorrect answers, and the model should recognize the correctness via verification questions. Then we test the model’s retraction performance as shown below.
>
> |  | Retraction Precision | Retraction Recall |
> | --- | --- | --- |
> | Llama3.1-8B-Instruct | 0.9012 | 0.2579 |
> | Llama3.1-70B-Instruct | 0.9126 | 0.5303 |
>
> From the results, we can see that the larger model exhibits better retraction performance with higher retraction precision and recall compared to smaller LLMs. However, its retraction performance remains far from ideal, with nearly half of the incorrect examples still not being retracted.
>
> We also reproduce the probing experiment on the 70B model to verify whether belief predicts retraction behavior in larger models. Given that Llama3.1-70B-Instruct has 80 layers and a hidden size of 8192 (Llama3.1-8B-Instruct: 32 layers, 4096-d), to mitigate overfitting we expand the UTQA probe-training set from 2400 to 8000 examples and train an independent *single-linear* probe at each layer. We find that Layers 29-35 are the most predictive layers for both factual correctness and retraction. The AUROC scores for these layers are shown below.
>
> |  | AUROC (retraction) | AUROC (factual correctness) | $\Delta$ |
> | --- | --- | --- | --- |
> | Layer 29 | 0.723 | 0.686 | 0.037 |
> | Layer 30 | 0.727 | 0.687 | 0.040 |
> | Layer 31 | 0.721 | 0.671 | 0.050 |
> | Layer 32 | 0.722 | 0.662 | 0.060 |
> | Layer 33 | 0.729 | 0.677 | 0.052 |
> | Layer 34 | 0.715 | 0.662 | 0.053 |
> | Layer 35 | 0.714 | 0.667 | 0.047 |
>
> We observe that the belief probes achieve moderately strong performance on predicting retraction behavior, and they consistently outperform prediction of factual correctness, which is consistent with our findings on smaller models across different model families. Their predictive power is somewhat lower than probes on smaller models, possibly because larger models are more expressive and encode multiple entangled features at the coarse layer level. More fine-grained extraction of belief representations, such as at the head level or using sparse autoencoders, may reveal more generalizable belief signals. Nevertheless, the overall trend indicates that **the retraction mechanism and its relationship to momentary belief remain consistent from smaller to larger LLMs across families.**
>
> ### Weakness 2 and Weakness 5: What Belief Probes Capture
>
> This is a highly valuable question. We fully agree that an interesting finding is that *the linear probe is highly predictive of whether the model will retract its answer*. In brief, this is because **the probes really captures the model’s *belief* in the answer, rather than the ground-truth correctness**. Below we give a more detailed explanation.
>
> Although the UTQA is labeled by factual correctness, what the probe learns from this dataset is the model’s *own internal* separation between what is correct and what is wrong. This distinction is not necessarily aligned with ground-truth correctness.
>
> A key asymmetry plausibly explains why this happens:
>
> - UTQA wrong answers are externally generated by GPT-4-turbo, and the model typically knows internally that these answers are wrong.
> - Continuation wrong answer are generated by the model itself.
>
> As a result,
>
> - On UTQA: belief $\approx$ factual correctness.
> - On continuation datasets: belief $\ne$ correctness, but belief $\to$ retraction.
>
> Our probing results highlight the gap between what is factually correct and what a model internally “believes” to be correct.
>
> Moreover, the probe’s belief signal aligns closely with observable retraction behavior, suggesting that retraction is a behavioral manifestation of this underlying internal representation. Our steering experiments further demonstrate that this link is causal.

---

> ### Author Response · Authors · 2025-11-21
> **[2/2] Response to Reviewer rHvH**
>
> ### Weakness 3: Group-Level Retraction Performance
>
> We compute group-level retraction rates by splitting examples into two groups: those the model retracts originally and those it does not, as you rightly suggested. Results for Llama3.1-8B-Instruct on Wikidata are shown below:
>
> |  | Retract Num | Total Num | Retraction Rate |
> | --- | --- | --- | --- |
> | Retract Group (belief-) | 163 | 172 | 0.9477 |
> | Non-retract Group (belief-) | 917 | 1030 | 0.8903 |
> | Retract Group (belief+) | 3 | 172 | 0.0174 |
> | Non-retract Group (belief+) | 1 | 1030 | 0.0010 |
>
> Negative belief steering substantially increases retraction in the original non-retract group, while positive belief steering greatly suppresses retraction in the original retract group. For the remaining two cases, the rates are close to 1 or 0, and the small differences can be attributed to randomness introduced by modifying hidden states through steering.
>
> These findings align with the overall changes in retraction rate shown in Figure 3.
>
> ### Weakness 4: Layer-Level Explanation
>
> Thank you for the sharp observation. The low probe performance in early layers is likely due to the fact that these layers primarily encode shallow lexical and syntactic features, rather than belief-related representations. Likewise, the final layers tend to emphasize surface-level generation and fluency, rather than deeper semantic reasoning. As a result, belief representations are most salient in the middle layers, where we observe the highest probe performance.
>
> ### Weakness 6: Contributions
>
> In this work, we apply established interpretability tools including linear probing, steering, and patching, to investigate a relatively underexplored question: why LLMs retract some wrong answers but not others. Specifically, we ask whether there exists an internal representation, independent of retraction labels, that nonetheless governs external retraction behavior.
>
> We hope our analysis contributes to a deeper understanding of the retraction mechanisms in LLMs, offering not only evaluation results but also insights into how LLMs internally decide to retract and how supervised fine-tuning relates to internal representations.
>
> ---
> We hope our responses have addressed your concerns. We would be delighted to engage in further discussion and thank you again for your constructive feedback.

---

### Official Review · Reviewer_vuwy · 2025-11-01

**Soundness:** 2
**Presentation:** 3
**Contribution:** 2
**Rating:** 4
**Confidence:** 4

**Summary:**

This paper studies spontaneous retraction—cases where an LLM, after giving an answer, immediately acknowledges it is wrong without being prompted. The authors build model-specific “continuation” testbeds from two knowledge tasks (WIKIDATA; CELEBRITY) and evaluate three 7–8B instruction-tuned models. They find 1. models can retract but do so rarely (low recall), even when separate verification questions show the model “knows” the answer is wrong; 2. a linear probe trained on external true/false datasets provides a belief signal that predicts retraction much better than correctness; 3. activation steering along belief directions causally modulates retraction (negative steering → frequent retraction; positive steering → almost none); and 4. SFT improves in-distribution retraction and aligns internal belief more closely with factual correctness.

**Strengths:**

- Clear, focused problem: Spontaneous retraction is practically relevant and distinct from multi-turn self-correction.

- Neat empirical finding: Linear probes of hidden states correlate strongly with retraction but less with ground-truth correctness, clarifying what such probes actually capture.

- Causality evidence: Activation steering gives credible leverage that belief directions are not merely correlates but drivers of retraction behavior.

- Mechanistic analysis: Patching experiments suggest attention value vectors (not only attention weights) are a primary pathway by which belief influences retraction.

- Training connection: Showing that SFT improves retraction by aligning internal belief bridges interpretability results with standard training practice.

**Weaknesses:**

- **External validity / scope** All core experiments use three small instruction models and two knowledge-centric datasets. It’s unclear if belief→retraction generalizes to larger or reasoning-style models, to non-factoid tasks, or to tool-augmented settings.

- **Evaluation dependence on an LLM judge** Retraction is judged automatically (Llama-3.3-70B). Although convenient, relying on a single judge risks systematic bias and false positives/negatives (e.g., hedged text vs true retraction). Human audits or multi-judge agreement would strengthen claims.

- **Prompt / decoding sensitivity** The phenomenon may hinge on stopping behavior and continuation prompts (the “is-appended” setting materially changes results). More thorough robustness to decoding parameters, prompt templates, and stop-criteria would help.

- **Probe training / generalization** Belief probes trained on UTQA then applied OOD to continuation data work well for retraction but less for correctness. This invites concerns about dataset-specific artifacts (style, length, lexical cues). More ablations on token position, layer selection, and context length would clarify.

- **Steering hyperparameters** Steering strength/layer ranges are manually searched. The narrative would benefit from systematic sweeps with confidence intervals and controls for side effects (fluency, length, off-topic drift).

- **Statistical reporting** Many results are presented as point metrics. Please include CIs, random-seed variance, and significance tests for retraction precision/recall and AUROC curves.

**Questions:**

- How consistent are retraction labels across multiple judges or human raters? Any inter-annotator stats?

- How does temperature, top-p, and max-tokens affect retraction rates independently of belief steering?

- Can you quantify textual markers of retraction (e.g., “I was wrong,” “Correction: …”) and report precision/recall for each subtype?

- Do belief directions found on one model family transfer to another? Any cross-model steering experiments?

- Beyond attention values, did you test MLP value patching or logit lens analyses to localize belief-to-retraction pathways?

---

> ### Author Response · Authors · 2025-11-21
> **[1/4] Response to Reviewer vuwy**
>
> Thank you for recognizing our contribution to understanding retraction behaviors of LLMs and further bridging interpretability results with standard training practice! Here are the responses to your concerns.
>
> ### Question 1 and Weakness 2: LLM Judge Reliability
>
> As described in Appendix B.6, we have manually evaluated 40 randomly selected examples for each judge prompt. Llama3.3-70B-Instruct shows high agreement with human judgment, achieving 39/40 on Wikidata and 38/40 on Celebrity.
>
> To further validate judge reliability, we use GPT-4.1-mini to evaluate Llama3.1-8B-Instruct’s retraction performance under no-steering, positive-steering, and negative-steering settings (1202 * 3 = 3606 examples on Wikidata; 826 * 3 = 2478 examples on Celebirty). We then compute agreement between GPT-4.1-mini and Llama3.3-70B-Instruct:
>
> |  | Consistency | Cohen Kappa |
> | --- | --- | --- |
> | Wikidata | 0.9750 | 0.9445 |
> | Celebrity | 0.9730 | 0.9364 |
>
> Llama3.3-70B-Instruct achieves very high agreement with both human annotators and another LLM judge, while being convenient and cost-free.
>
> ### Question 2 and Weakness 3: Prompt and Decoding Sensitivity
>
> To examine how prompt templates and decoding hyperparameters influence the model’s original retraction performance, and whether belief still causally affects retraction, we conduct experiments using various prompt formulations and decoding settings. In brief, although both prompt templates and decoding hyperparameters do affect the original retraction performance, the model still does not retract enough, and belief steering works in all scenarios. All experiments below use Llama3.1-8B-Instruct and Wikidata.
>
> **1. Prompt Sensitivity**
>
> We test three prompt templates:
>
> **original_prompt:** Name an association football player who was born in Naples, Italy.
>
> **new_prompt1:** Can you name an association football player who was born in Naples, Italy?
>
> **new_prompt2:** Which association football player was born in Naples, Italy? Just name one.
>
> **adv_prompt:** Barack Obama is a politician who was born in New York City, United States. Name an association football player who was born in Naples, Italy. (Adversarial prompt template: it adds an incorrect statement before the question to potentially bias the model’s belief.)
>
> Results are shown below:
>
> | new_prompt1 | Retraction Precision | Retraction Recall |
> | --- | --- | --- |
> | No Steering | 0.8994 | 0.4609 |
> | Belief− Steering | 0.5296 | 0.8935 |
> | Belief+ Steering | 0.9412 | 0.0266 |
>
> | new_prompt2 | Retraction Precision | Retraction Recall |
> | --- | --- | --- |
> | No Steering | 0.9211 | 0.0582 |
> | Belief− Steering | 0.5115 | 0.8902 |
> | Belief+ Steering | 1.0 | 0.0017 |
>
> | adv_prompt | Retraction Precision | Retraction Recall |
> | --- | --- | --- |
> | No Steering | 0.8138 | 0.4725 |
> | Belief− Steering | 0.5280 | 0.8319 |
> | Belief+ Steering | 0.9091 | 0.0166 |
>
> Consistent with the observations in Section 4.2, negative belief steering (i.e., belief- steering) leads the model to retract significantly more often, while positive belief steering (i.e., belief+ steering) causes the model to almost never retract. This demonstrates that **belief steering is robust to changes in prompt phrasing.**
>
> **2. Decoding Sensitivity**
>
> In this paper, we use greedy decoding for determinism. Here, we supplement the analysis with temperature sampling (temperature=0.7, top-p=0.95), repeated three times with different random seeds. Numbers in parentheses indicate the standard deviation.
>
> |  | Retraction Precision | Retraction Recall |
> | --- | --- | --- |
> | Greedy Decoding | 0.9012 | 0.2579 |
> | Temperature Sampling | 0.8814 (0.0105) | 0.3128 (0.0088) |
> | Temperature Sampling with Belief- Steering | 0.5257 (0.0013) | 0.8980 (0.0101) |
> | Temperature Sampling with Belief+ Steering | 0.9551 (0.0024) | 0.0355 (0.0019) |
>
> These results show that **the link between belief and retraction remains stable under stochastic decoding.**
>
> Lastly, for the decoding hyperparameter `max_new_tokens`, we set it to 128 to give the model sufficient space to produce a retraction. Using a smaller value or other stop criteria may truncate the model’s response early, which would unfairly reduce the measured retraction rate.

---

> ### Author Response · Authors · 2025-11-21
> **[2/4] Response to Reviewer vuwy**
>
> ### Question 3: Textual Markers of Retraction
>
> We randomly sample 100 continuation responses from Llama3.1-8B-Instruct on Wikidata and manually annotate the textual markers of retraction. The statistics are summarized below:
>
> |  | Num |
> | --- | --- |
> | different condition | 46 |
> | be not | 25 |
> | ‘s | 17 |
> | is not the answer | 6 |
> | is unknown | 5 |
>
> The top 3 types are illustrated below:
>
> - different condition: “was born Ogidi, not Lagos”.
> - be not: “was not born in Maxico City, Mexico”; unlike *different condition*, the model may not know the correct birthplace but recognizes that the one in the prompt is incorrect.
> - ‘s: “’s sister”, where the model shifts the entity being discussed.
>
> Since simple string matching may introduce false positives or false negatives, we use LLM-as-a-Judge for more accurate and flexible evaluation. When textual markers of retraction are present, the case is categorized as a retraction; in such cases, the retraction rate is 1, and retraction precision or recall is not meaningful.
>
> ### Question 4: Cross-Model Belief Vectors
>
> Thank you for this thoughtful question. We agree that cross-model steering is an interesting direction. In this work, we focus on within-model causal analysis, because belief vectors depend on model-specific layer geometry and hidden dimensionality, making direct transfer non-trivial without an alignment mapping. For example, Llama3.1-8B-Instruct (4096-d) and Qwen2.5-7B-Instruct (3584-d) differ both in dimensionality and in where belief is represented across layers as shown in Figure 3(a) and Figure 5.
>
> While we did not perform cross-model steering experiments due to these architectural mismatches, we did evaluate the phenomenon across multiple model families (Llama, Qwen, Olmo) and found that each model independently exhibits the same underlying retraction mechanism.
>
> ### Question 5: MLP Value Patching and Logit Lens
>
> Our motivation for testing the attention mechanism comes from a hypothesis that *models fail to retract when they do not have sufficiently attend to the given answer*, as described in Lines 370-371. We then measure whether attention weights or attention value vectors that encode information play a more important role.
>
> MLP activations and attention value vectors are intertwined and can both be interpreted as modifying the information the answer brings. For this reason, we did not include MLP patching in the paper. However, motivated by your comment, we additionally patch the MLP vectors at all layers.
>
> |  | Retraction Precision | Retraction Recall | Retraction Precision (is-appended) | Retraction Recall (is-appended) |
> | --- | --- | --- | --- | --- |
> | No Steer | 0.9012 | 0.2579 | 0.8254 | 0.5740 |
> | Patch Attention Value Vectors (belief-) | 0.5249 | 0.5441 | 0.5069 | 0.9784 |
> | Patch MLP (belief-) | 0.6436 | 0.7121 | 0.6380 | 0.7271 |
> | Full Steer (belief-) | 0.5157 | 0.9268 | 0.5026 | 0.9717 |
> | Patch Attention Value Vectors (belief+) | 0.9700 | 0.1614 | 0.9851 | 0.3311 |
> | Patch MLP (belief+) | 1.0000 | 0.0116 | 0.8865 | 0.4809 |
> | Full Steer (belief+) | 1.0000 | 0.0067 | 0.9847 | 0.3211 |
>
> As expected, without the appended *is*, patching the MLP recovers more retraction performance because the modification is written directly into the residual stream. In the *is*-appended setting, patching attention value vectors is more effective because it directly affects later-token generation. However, patching MLP also works since changing the MLP output alters the attention value vectors in subsequent layers along the information flow path.
>
> For the logit lens analysis, since its interpretability strongly depends on the specific observed token, we conduct a preliminary experiment examining the change in logits for the token “.” and “EOS” with (layer 6-14) and without steering. We find that negative belief steering suppresses the generation of “.” as early as layer 6 and suppresses “EOS” starting from layer 12, with the degree of suppression increasing gradually. This aligns with the decreased stop-rate phenomenon we report in Table 3. Conversely, positive belief steering increases the likelihood of generating these two tokens in a similar trend. These observations suggest that belief steering modifies next-token predictions not only at the final layers.

---

> ### Author Response · Authors · 2025-11-21
> **[3/4] Response to Reviewer vuwy**
>
> ### Weakness 3: External Validity / Scope
>
> We appreciate your feedback regarding the robustness of the belief $\to$ retraction relationship. To further validate this mechanism, we additionally conduct experiments on a larger model (Llama3.1-70B-Instruct) as well as on a reasoning-intensive task (GSM8k). Both experiments support the same conclusion, suggesting that this belief-driven retraction mechanism is consistent across model scales and task types.
>
> **1. Larger LLMs**
>
> We evaluate Llama3.1-70B-Instruct to examine whether our findings hold for larger LLMs. We construct its Wikidata continuation test set of 4492 examples with balanced correct and wrong answers, and test that its retraction precision is 0.9126, and retraction recall is 0.5303. Though the larger model exhibits better retraction performance than Llama3.1-8B-Instruct (precision: 0.9126 vs. 0.9012; recall: 0.5303 vs. 0.2579), there remains substantial room for improvement.
>
> To further verify whether belief predicts retraction behavior in larger models, we reproduce the probing experiment on the 70B model. Given that Llama3.1-70B-Instruct has 80 layers and a hidden size of 8192 (Llama3.1-8B-Instruct: 32 layers, 4096-d), to mitigate overfitting we expand the UTQA probe-training set from 2400 to 8000 examples and train an independent *single-linear* probe at each layer. We find that Layers 29-35 are the most predictive layers for both factual correctness and retraction. The AUROC scores for these layers are shown below.
>
> |  | AUROC (retraction) | AUROC (factual correctness) | $\Delta$ |
> | --- | --- | --- | --- |
> | Layer 29 | 0.723 | 0.686 | 0.037 |
> | Layer 30 | 0.727 | 0.687 | 0.040 |
> | Layer 31 | 0.721 | 0.671 | 0.050 |
> | Layer 32 | 0.722 | 0.662 | 0.060 |
> | Layer 33 | 0.729 | 0.677 | 0.052 |
> | Layer 34 | 0.715 | 0.662 | 0.053 |
> | Layer 35 | 0.714 | 0.667 | 0.047 |
>
> We observe that the belief probes achieve moderately strong performance on predicting retraction behavior, and they consistently outperform prediction of factual correctness, which is consistent with our findings on smaller models across different model families. Their predictive power is somewhat lower than probes on smaller models, possibly because larger models are more expressive and encode multiple entangled features at the coarse layer level. More fine-grained extraction of belief representations, such as at the head level or using sparse autoencoders, may reveal more generalizable belief signals. Nevertheless, the overall trend indicates that **the retraction mechanism and its relationship to momentary belief remain consistent from smaller to larger LLMs across families.**
>
> **2. Reasoning Tasks**
>
> We first collect Llama3.1-8B-Instruct’s trajectories on GSM8k that produce *incorrect* final answer via greedy decoding, and use GPT-4.1 to annotate the first incorrect token. This results in a total of 188 examples (where accuracy = 0 with no steering by construction). We then apply negative belief steering, without altering any other generation setting, and evaluate accuracy on this subset. To moderately amplify the effect, we steer layers 0-18 and intervene on the first incorrect token plus its preceding nine tokens (a randomly selected hyperparameter). Since math reasoning depends on multi-step computation, modifying early hidden states can propagate and correct later inference. We also report positive belief steering to reveal unintended effects.
>
> **Results:**
>
> |  | Acc |
> | --- | --- |
> | Belief- Steering | 37.77% |
> | Belief+ Steering | 17.55% |
>
> **Case Study:**
>
> > Question: Carla is downloading a 200 GB file. Normally she can download 2 GB/minute, but **40% of the way through the download, Windows forces a restart** to install updates, which takes 20 minutes. Then Carla has to restart the download from the beginning. How long does it take to download the file?
>
> No Steering: wrong
>
> > Assistant: ...3. After the restart, Carla has to download the remaining 60% of the file (100% - 40% = 60%)…
>
> Negative Belief Steering: correct *(steering takes effect on the italicized part)*
>
> > Assistant: ...3. After the restart, Carla has to download the remaining 60% *of the file, but Carla has to restart the download from the beginning. So Carla has to download the entire 200 GB again…*
>
> Positive Belief Steering: wrong *(steering takes effect on the italicized part)*
>
> > Assistant: ...3. After the restart, Carla has to download the remaining 60% *.*
> >
> >    *60% of 200 GB is 0.6 * 200 = 120 GB…*
>
> Recall that the belief vector is derived from a quite different dataset UTQA. Negative belief steering still activates retraction behavior in math reasoning and improve the final accuracy by around 20%, **demonstrating the robustness and generalization of our interpretability findings.**

---

> ### Author Response · Authors · 2025-11-21
> **[4/4] Response to Reviewer vuwy**
>
> ### Weakness 4: Probe Training / Generalization
>
> We would like to kindly clarify that “Belief probes trained on UTQA then applied OOD to continuation data work well for retraction but less for correctness” is exactly the phenomenon we aim to highlight, as reflected in you Strength 2:
>
> > Linear probes of hidden states correlate strongly with retraction but less with ground-truth correctness, clarifying what such probes actually capture.
>
> This also helps explain why these belief probes do not generalize well on out-of-distribution dataset as observed in prior work [1]: they capture the model’s own judgement about the truth of the world, which can be diverge from ground-truth correctness.
>
> Regarding experimental settings, we use the same token position for both the training and testing, that is, the last token of the answer span, as our analysis focuses on the model’s momentary belief about its generated answer and its subsequent retraction behavior. For layer selection, we report probes trained independently for every layer, providing a comprehensive view across the entire model. We would be happy to offer further clarification if you have additional concerns about these settings.
>
> [1] Are the Hidden States Hiding Something? Testing the Limits of Factuality-Encoding Capabilities in LLMs.
>
> ### Weakness 5: Steering Hyperparameters
>
> Thank you for the suggestion. Following prior work [1] on activation steering, we perform a targeted manual search over steering hyperparameters using a small validation set, and ensure that they do not negatively affect generation quality, as detailed in Appendix B.3. While more exhaustive hyperparameter sweeps could potentially identify even better settings, we find that the current choices produce stable and consistent effects across different datasets. We will clarify this justification in the revised version.
>
> [1] Programming Refusal with Conditional Activation Steering.
>
> ### Weakness 6: Statistical Reporting
>
> Thank you for raising this point. We have updated the probing results in the PDF by running each experiment three times and including the standard deviation in Figure 2 and 4. With these new results, and given that all probes are trained independently for each layer, we believe the findings are robust. Additionally, we use greedy decoding for all inference in the paper to ensure determinism, so there is no variance for generation results introduced by random seeds.
>
> ---
> We hope our responses have adequately resolved your concerns. We sincerely appreciate your constructive feedback and would be happy to continue the discussion.

---

### Official Review · Reviewer_eA1T · 2025-11-01

**Soundness:** 2
**Presentation:** 2
**Contribution:** 2
**Rating:** 4
**Confidence:** 3

**Summary:**

This paper proposes a reliable predictor of retraction: the model’s momentary belief, as measured by a probe on its internal states that is trained to predict correctness on external datasets unrelated to retraction. Supervised fine-tuning shows to improve retraction performance by helping a model learn more accurate internal belief.

**Strengths:**

1. Regarding originality, momentary belief appears to be an interesting concept. See questions/weaknesses below so I can better judge the novelty.

2. I like the supervised fine-tuning part, which connects the idea of LLMs' internal belief.

**Weaknesses:**

1. Soundness needs to be improved by running experiments on larger models. The largest model studied is 8B, so I am not sure how the findings generalize to larger models. It is necessary to discuss the similarity and difference between large and smaller models in terms of your momentary belief concept.

2. I am not sure the utility of momentary belief. What are the final metrics of measuring the utility of momentary belief? The clarity needs to be improved.

3. I am not sure the efficiency of computing momentary belief. If your concept of momentary belief is indeed helpful, it is necessary to show the comparison on performance, computation costs, and inference latency across a wide range of baselines.

4. Writing needs to be improved in order to reach a publishable state. For example, I am not sure how your definition of "retraction" differs from "backtracking" as the question asked below. This is my concern, because I use it to judge the novelty of this paper.

**Questions:**

1. In your abstract, can you explain "Steering experiments further demonstrate that model belief causally drives retraction"? You used the word "causally" throughout your paper, but I still find a hard time understanding it.

2. Since you define retraction in Section 3.1, is it the same as "backtracking" that is more commonly used in other literature such as [1]?

[1] "To Backtrack or Not to Backtrack: When Sequential Search Limits Model Reasoning"

---

> ### Author Response · Authors · 2025-11-21
> **[1/2] Response to Reviewer eA1T**
>
> Thank you for devoting your time and effort to reviewing our paper. Below, we respond to your concerns point by point.
>
> ### Question 1: The Meaning of “Causally”
>
> Thank you for your question. We use the term “causally” because our steering experiments perform interventions on the model’s internal belief representation, rather than merely observing correlations. Specifically, we directly modify the hidden states along the belief direction at test time. When we increase the belief value, the model retracts significantly less, and when we decrease it, the model retracts substantially more, while all other inputs and decoding settings are kept identical.
>
> Since the only manipulated variable is the internal belief activation, and this manipulation reliably changes retraction behavior, the experiment provides *causal* evidence that the belief representation drives retraction.
>
> ### Question 2 and Weakness 4: Retraction vs. Backtracking
>
> Thank you for pointing us to the backtracking paper [1]. In [1], backtracking refers to a search-based correction mechanism that arises during *multi-step reasoning*: the model explores a branch, detects inconsistency, and then explicitly returns to an earlier node to pursue a different path, often instantiated through tree-structured decoding or DFS-style search.
>
> In contrast, our definition of “retraction” captures a different phenomenon: the model produces an incorrect answer and then immediately revises or negates that answer, **without invoking a search process or revisiting earlier branches.**
>
> A reasoning example of retraction from GSM8k is like:
>
> > User: Carla is downloading a 200 GB file. Normally she can download 2 GB/minute, but **40% of the way through the download, Windows forces a restart** to install updates, which takes 20 minutes. Then Carla has to restart the download from the beginning. How long does it take to download the file?
>
> > Assistant: …3. After the restart, Carla has to download the remaining 60% of the file, **but** Carla has to restart the download from the beginning. So Carla has to download the entire 200 GB again…
>
> Here, the model states an incorrect assumption (”remaining 60%”), instantly negates it, and replaces it with the corrected statement (”restart from the beginning”). This behavior does not involve exploring alternative branches or returning to an earlier reasoning node, which distinguishes it clearly from the backtracking mechanism described in [1].
>
> Additionally, our paper focuses not on improving performance via this behavior we study, but rather on constructing an evaluation framework and *analyzing when and why retraction occurs through the lens of internal representations.*
>
> [1] To Backtrack or Not to Backtrack: When Sequential Search Limits Model Reasoning.

---

> ### Author Response · Authors · 2025-11-21
> **[2/2] Response to Reviewer eA1T**
>
> ### Weakness 1: Larger LLMs
>
> Motivated by your comment, we additionally evaluate Llama3.1-70B-Instruct to examine whether our findings hold for larger LLMs. We construct its Wikidata continuation test set of 4492 examples with balanced correct and wrong answers, and test that its retraction precision is 0.9126, and retraction recall is 0.5303. Though the larger model exhibits better retraction performance than Llama3.1-8B-Instruct (precision: 0.9126 vs. 0.9012; recall: 0.5303 vs. 0.2579), there remains substantial room for improvement.
>
> To further verify whether momentary belief predicts retraction behavior in larger models, we reproduce the probing experiment on the 70B model. Given that Llama3.1-70B-Instruct has 80 layers and a hidden size of 8192 (Llama3.1-8B-Instruct: 32 layers, 4096-d), to mitigate overfitting we expand the UTQA probe-training set from 2400 to 8000 examples and train an independent *single-linear* probe at each layer. We find that Layers 29-35 are the most predictive layers for both factual correctness and retraction. The AUROC scores for these layers are shown below.
>
> |  | AUROC (retraction) | AUROC (factual correctness) | $\Delta$ |
> | --- | --- | --- | --- |
> | Layer 29 | 0.723 | 0.686 | 0.037 |
> | Layer 30 | 0.727 | 0.687 | 0.040 |
> | Layer 31 | 0.721 | 0.671 | 0.050 |
> | Layer 32 | 0.722 | 0.662 | 0.060 |
> | Layer 33 | 0.729 | 0.677 | 0.052 |
> | Layer 34 | 0.715 | 0.662 | 0.053 |
> | Layer 35 | 0.714 | 0.667 | 0.047 |
>
> We observe that the belief probes achieve moderately strong performance on predicting retraction behavior, and they consistently outperform prediction of factual correctness, which aligns with our findings on smaller models across different model families. Their predictive power is somewhat lower than probes on smaller models, possibly because larger models are more expressive and encode multiple entangled features at the coarse layer level. More fine-grained extraction of belief representations, such as at the head level or using sparse autoencoders, may reveal more generalizable belief signals. Nevertheless, the overall trend indicates that **the retraction mechanism and its relationship to momentary belief remain consistent from smaller to larger LLMs across families.**
>
> ### Weakness 2 and 3: The Function of Belief
>
> Thank you for your deep thought on the practical utility and efficiency of momentary belief, but we would like to kindly clarify the focus of our paper.
>
> In this work, our primary goal is to investigate when and why LLMs retract, and to provide a more reliable internal explanation for this external behavior.
>
> Although we use the term “momentary belief” instead of “truthfulness” in this paper to distinguish it from world-knowledge factuality, this concept itself originates from prior work [1,2], as detailed in Lines 066-070. It has been demonstrated that a model has its internal belief of what is correct and what is wrong. This motivates two possible interpretations of retraction: (1) Models may internally believe that their wrong answers are true, which causes them to not retract, or (2) Models may recognize their answers to be false, yet still choose not to verbalize this belief.
>
> Through probing, steering, and SFT experiments, we connect internal belief to external retraction behavior, thereby improving our understanding of model mechanisms rather than treating models as uncontrollable black boxes.
>
> Therefore, in this paper, we focus on the *explanatory value* of momentary belief, not performance enhancement, and therefore do not include utility/efficiency comparisons.
>
> [1] Inference-Time Intervention: Eliciting Truthful Answers from a Language Model.
>
> [2] On the Universal Truthfulness Hyperplane Inside LLMs.
>
> ---
> Thank you again for your detailed and thoughtful feedback. We hope our responses have addressed your concerns. Please don’t hesitate to let us know if any further clarification would be helpful.

---

### Author Response · Authors · 2025-12-01
**Final Summary for AC Review**

As the discussion period closed earlier than expected, we unfortunately did not have the opportunity to receive any reviewer feedback on our rebuttal. Nonetheless, we believe that our responses and additional experiments have addressed the main concerns raised in review, and we provide a brief summary for the AC’s convenience.

1. We add new evaluations on a **larger model** (Llama3.1-70B-Instruct), confirming that internal belief remains predictive of retraction at scale (Appendix C.1.1).
2. We conduct robustness studies under **prompt and decoding variations**, showing the belief-retraction causality is stable beyond the original experimental setup (Appendix C.3).
3. We extend analysis to **math reasoning** (GSM8k), where negative belief steering improves accuracy by ~20%, highlighting generalization beyond factoid QA (Appendix C.7).
4. We validate the reliability of our LLM-as-a-Judge setting using GPT-4.1-mini in addition to human annotation in Appendix B.6, observing high agreement.

We also responded to reviewer-specific questions:

1. For **Reviewer eA1T**, we clarify the meaning of “causally” and distinguish the behavior of retraction from backtracking.
2. For **Reviewer vuwy**, we add statistical reporting (Figure 2 and 4), justify steering hyperparameters (Appendix B.3), and discuss potential extensions including cross-model steering and MLP value patching.
3. For **Reviewer rHvH**, we clarify that our contribution is identifying whether an internal representation, independent of retraction labels, can nonetheless govern external retraction behavior, using established interpretability tools to reveal this previously underexplored mechanism.
4. For **Reviewer VBRM**, we engage with questions on probe complexity and cross-model steering.

Overall, reviewers recognized the value of formalizing spontaneous retraction and the contribution of uncovering a consistent mechanistic link between internal belief and external retraction behavior. The connection to supervised fine-tuning was also positively received. We believe our work contributes to a deeper understanding of the retraction mechanisms in LLMs, offering not only evaluation results but also insights into how LLMs internally decide to retract and how supervised fine-tuning relates to internal representations.

We appreciate all the reviewers’ time and constructive feedback, and sincerely thank the AC for considering our rebuttal during the final decision process.

---

### Note · Authors · 2026-01-05

I have read and agree with the venue's withdrawal policy on behalf of myself and my co-authors.